# Towards Automatic Segmentation and Recognition of Multiple Precast Concrete Elements in Outdoor Laser Scan Data

**Jiepeng Liu [1], Dongsheng Li [1] ⓘ, Liang Feng [2],\* ⓘ, Pengkun Liu [1] and Wenbo Wu [2]**

[1] School of Civil Engineering, Chongqing University, Chongqing 400045, China; liujp@cqu.edu.cn (J.L.); lds@cqu.edu.cn (D.L.); pengkunliu@cqu.edu.cn (P.L.)

[2] College of Computer Science, Chongqing University, Chongqing 400045, China; wuwenbo@cqu.edu.cn

\* Correspondence: liangf@cqu.edu.cn

**Abstract:** To date, to improve construction quality and efficiency and reduce environmental pollution, the use of precast concrete elements (PCEs) has become popular in civil engineering. As PCEs are manufactured in a batch manner and possess complicated shapes, traditional manual inspection methods cannot meet today's requirements in terms of production rate of PCEs. The manual inspection of PCEs needs to be conducted one by one after the production, resulting in the excessive storage of finished PCEs in the storage yards. Therefore, many studies have proposed the use of terrestrial laser scanners (TLSs) for the quality inspection of PCEs. However, all these studies focus on the data of a single PCE or a single surface of PCE, which is acquired from a unique or predefined scanning angle. It is thus still inefficient and impractical in reality, where hundred types of PCEs with different properties may exist. Taking this cue, this study proposes to scan multiple PCEs simultaneously to improve the inspection efficiency by using TLSs. In particular, a segmentation and recognition approach is proposed to automatically extract and identify the different types of PCEs in a large amount of outdoor laser scan data. For the data segmentation, 3D data is first converted into 2D images. Image processing is then combined with radially bounded nearest neighbor graph (RBNN) algorithm to speed up the laser scan data segmentation. For the PCE recognition, based on the as-designed models of PCEs in building information modeling (BIM), the proposed method uses a coarse matching and a fine matching to recognize the type of each PCE data. To the best of our knowledge, no research work has been conducted on the automatic recognition of PCEs from a million or even ten million of the outdoor laser scan points, which contain many different types of PCEs. To verify the feasibility of the proposed method, experimental studies have been conducted on the PCE outdoor laser scan data, considering the shape, type, and amount of PCEs. In total, 22 PCEs including 12 different types are involved in this paper. Experiment results confirm the effectiveness and efficiency of the proposed approach for automatic segmentation and recognition of different PCEs.

**Keywords:** image segmentation; automatic segmentation and recognition; as-designed model; outdoor laser scan data; precast concrete elements

## 1. Introduction

Over recent decades, with the vigorous development of prefabricated construction, precast concrete elements (PCEs) have been popularly used in civil engineering. In contrast to traditional cast-in-situ concrete components, PCEs have many advantages with respect to construction efficiency, construction cost, working, and manufacturing environment [1–3]. On one hand, due to

the off-site fabrication, PCEs are manufactured efficiently in a batch manner at the prefabrication workshop, in parallel to the site preparation activities. For example, PCEs with regular geometries, such as precast concrete (PC) superimposed slabs (Figure 1a), are produced by pipelining. In addition, big PCEs with irregular geometries, such as decorative PC column, can be quickly manufactured through the standard steel molds (Figure 1b). On the other hand, PCEs satisfy the requirements in the design stage better than the cast-in-situ concrete components. This is because the prefabricated plant can easily meet the curing requirements of the concrete by controlling temperature and humidity of the environment. As the use of PCEs becomes widespread, it is worth noting that successful prefabricated construction depends on a strict control of dimensional tolerances [4–7]. However, as shown in Table 1 according to the tolerance manual for precast and pre-stressed concrete in Chinese codes [8,9], the current main quality inspection methods of PCEs are manual inspection using rulers or measurement tapes, which is time-consuming, costly, and labor intensive. Due to the complicated shapes and large amount of PCEs in production, manual inspection methods are unable to catch up with the manufacturing efficiency of PCEs in the prefabrication workshop. Before being transported to the site for installation, all types of produced PCEs thus will be sent to an open yard for inspection and storage. Therefore, there is an increasing need to propose a technique that can adapt to this open environment for efficient quality inspection of PCEs.

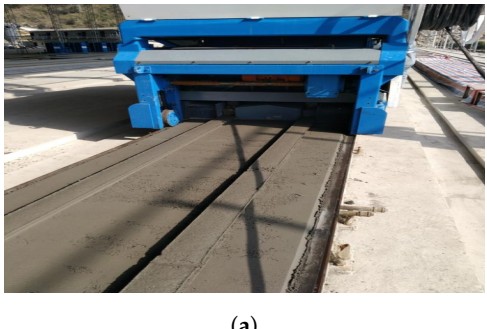
(**a**)
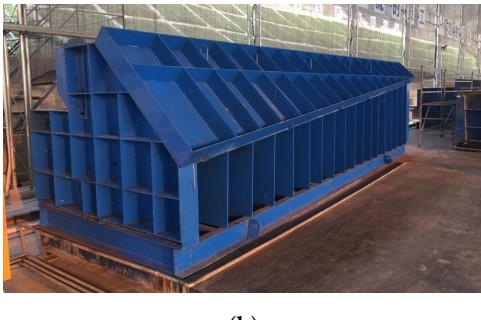
(**b**)

**Figure 1.** Efficient production of precast concrete element: (**a**) Continuous fabrication of PC superimposed slab; (**b**) Reusable steel mold of decorative PC column.

**Table 1.** Main dimensional tolerances and test methods for PCEs.

| Project | | | Tolerances (mm) | Measurement Methods |
|---|---|---|---|---|
| Length | Beam, Column, Slab, Truss | <12 m | ±5 | Rule detection |
| | | ≥12 m and <18 m | ±10 | |
| | | ≥18 m | ±20 | |
| | Wall panel | | ±4 | |
| Width, Height (Thickness) | The section of Beam, Column, Slab, Truss | | ±5 | Rule detection |
| | Wall panel | | ±3 | |

To improve the traditional manual measurement methods, researchers have proposed the non-contact measurement methods with the aid of auxiliary devices [10]. The image-based methods and 3D laser scanning-based methods have been widely accepted. On one hand, the image-based methods are characterized by the rapid processing speed and low-cost consumption. They have been used to evaluate the concrete surface quality, such as detecting cracks or air pockets [11,12]. However, as shown in Table 1, the dimensions of the PCEs need to be controlled within several millimeters in the relevant specifications. For the PCEs less than 12m in length, the maximum allowable tolerance is only ±5 mm. The accuracy of the image-based methods for estimating the dimensional error cannot meet the tolerance requirements of PCEs. This is because the image-based methods could be significantly affected by the quality of the images and the lighting conditions for collecting data [13]. On the other hand, taking advantage of the spacious environment of the storage yard, the high-tech field

data acquisition system (e.g., 3D laser scanner) is employed to obtain the geometric information of PCEs, which has the advantages of good ranging error (typically 2∼6 mm at 50 m), low ranging noise (typically 0.3∼0.4 mm within 10 m) and high measurement speed (up to 976,000 points/s). Usually, laser scan data obtained by commercial laser scanners contains not only the distance measurement of scanned objects, but also the red, green and blue (RGB) values of each point. Each point is represented as a six-dimensional array $(x, y, z, r, g, b)$. These RGB values are calculated by the reflected signals from the scanned objects [14]. Based on the obtained laser scan data, the quality inspection of PCEs can be conducted. For instances, Kim et al. [4] developed a method using an edge and corner extraction technique to estimate the dimensional properties of PC panels in laboratory size. In order to verify the effectiveness of this method, Kim et al. [5] set a 3D laser scanner directly above full-scale PC panels at the storage yard for data acquisition. An improved vector sum methods proposed in reference [4] was employed to perform dimensional quality inspection. Based on $F_F$ number, Wang et al. [15] proposed a technique to estimate the surface flatness of PC panels by scanning above these panels. They evaluated the distortion of PC panel according to the warping of each corner, the bowing of each edge and the differential elevation between adjacent points. In order to perform the quality inspection of PC panels with irregular geometry, Wang et al. [6] developed a technique using coordinate transformation and inner corner extraction to estimate the quality of shear key or flat duct in a PC panel. Yoon et al. [16] developed an approach to identify the optimal position of precast bridge deck slabs with respect to precast girders by solving a nonlinear minimization problem, which considered the deck slabs and sizes of the extracted PCE. Furthermore, Wang et al. [7] proposed to estimate the dimensions of a full-scale PC bridge deck panel by considering an optimal scan scheme and a shear pockets extraction method. Reverse modeling of the scanned PC bridge deck panel was realized in BIM system. However, notably, these efforts mainly focus on dimensional quality assessment methods of a single object data. In particular, all these methods can only work with the data of a single PCE with unique shape, or a single surface of PCE, which are captured from a unique or predefined scanning angle (e.g., directly above the PCE). As aforementioned, PCEs are produced in a batch manner. The inspection efficiency of the one-by-one scanning is still inadequate for the cases containing multiple types of PCEs possessing different shapes. Therefore, considering the open environment of the storage yard, there is an enough area that is available to scan multiple PCEs simultaneously. Due to the panoramic scanning capability of the 3D laser scanner, there is no doubt that the inspection efficiency will has a significant improvement by replacing the one-by-one scanning mode with the scanning of multiple PCEs simultaneously. Furthermore, after obtaining the total scan data including different types of PCEs, the data segmentation and recognition of these PCEs are critical for the later quality inspection. No research work has been conducted on the automatic segmentation and recognition of PCEs from these outdoor laser scan data, which contains many different types of PCEs. This work thus presents the first attempt to fill this gap.

To process the scanned multiple PCE data, the main challenge is to extract each PCE data from the huge amount of noisy outdoor laser scan data, and then accurately recognize the different types of PCEs, which can thereby be easily stored in correspondence with the as-designed model in BIM. Specifically, to ensure the accuracy of PCE quality inspection, a high scan density is required for the PCE scan, which leads to a huge amount of laser scan data. To avoid errors caused by data sampling, the PCE data is directly extracted from the original data. This will further result in a huge computational burden. Next, high complexity exists in the outdoor laser scanner data. In particular, the total data acquired may include the noisy background and multiple PCEs. Due to the variety of produced PCEs, it is important to recognize the type of each PCE. Keeping the above in mind, in this paper, we choose the TLS with built-in camera to scan these PCEs. Instead of dealing with the complex laser scan data directly, we propose to convert the 3D data into 2D images, and perform automatic segmentation of the outdoor laser scan data in 2D images. In this way, the success achieved in 2D image processing [11,12] can be leveraged for the complex PCE segmentation with acceptable computational

cost. Moreover, taking the as-designed models in BIM into account [13,17,18], it provides reference object that can be used for accurate PCE recognition.

In particular, this study aims to propose an automatic segmentation and recognition method to identify multiple PCEs simultaneously in outdoor laser scan data, thereby to improve the inspection efficiency of PCEs. For the data segmentation, 3D data is first converted into 2D images based on the RGB values of the laser scan data. The image processing technology such as clustering and edge recognition for image segmentation is then used to process the obtained images. Next, a novel active window method is developed to quickly extract the data within segmented image clusters based on the edge image. To avoid under-segmentation, the RBNN algorithm [19] is then used to further separate the data obtained by image segmentation. Lastly, a reconstruction of segments is used to avoid over-segmentation. For PCE recognition, an analysis of the normal vector distribution is used to recognize the PCEs. The accurate identification is achieved by coarse and fine matching based on as-designed models in BIM. Last but not the least, to verify the validity of the proposed approach, experiments on outdoor laser scan data which includes multiple PCEs have been conducted with respect to three aspects, which are PCE shape, PCE type, and PCE amount. The comprehensive experimental data includes a total of 22 PCEs with 12 types, which are given as follows:

- For PCE shape, we scan 3 PCEs of the same type but with different shapes simultaneously.
- For PCE type, we scan 3 PCEs of the different types simultaneously.
- For PCE amount, we scan 16 PCEs simultaneously, which contain 9 different types of PCEs.

The contribution of the present study can then be summarized as follows:

- To improve the quality inspection efficiency of PCEs, we have proposed an automatic segmentation and recognition approach to extract and identify the accurate type of each PCE in outdoor laser scan data. To the best of our knowledge, this is the first attempt in the literature for automatic recognition of multiple PCEs scanned simultaneously.
- To handle the huge computation burden, we have proposed an approach based on the image processing and RBNN algorithm to segment the outdoor laser scan data.
- To solve the problem of backtracking from 2D image cluster to 3D laser scan data, we have developed a novel algorithm using active window to trace back the laser scan data based on an edge image.
- To verify the effectiveness of the proposed approach, experiments on outdoor laser scan data containing multiple PCEs have been performed in three aspects. To the best of our knowledge, no research has investigated such comprehensive experiments before.

The rest of this paper is organized as follows: Section 2 provides a literature review of the segmentation and identification of laser scan data. Section 3 presents the details of the automatic type identification for PCEs. In Section 4, the effectiveness of the proposed approach is experimentally validated. Lastly, the conclusions of this study are given in Section 5.

## 2. Preliminary

In this section, the research background and the motivation of this study are discussed introduced in Section 2.1. Next, the literature review of related work is given in Section 2.2.

### 2.1. Research Background & Motivation

As aforementioned, since the production of PCE is in a batch manner, the quality inspection of each produced PCE cannot be achieved immediately by the tedious manual methods. Although the 3D scanning technology has a significant improvement of data acquisition, the existing quality inspection techniques based on laser scan data are still unable to compensate the gap between the efficiency of production and quality inspection, since existing methods have PCE-specific requirements on data acquisition, and cannot deal with a batch of PCEs possessing different properties, such as shape,

volume, and type. Therefore, PCEs that have not been transported to the field are stacked in the yard. At the storage yards served for high PCE output, it is common to see that the produced PCEs are evenly lined along the sides of the various access routes [20], waiting for quality inspection (Figure 2).

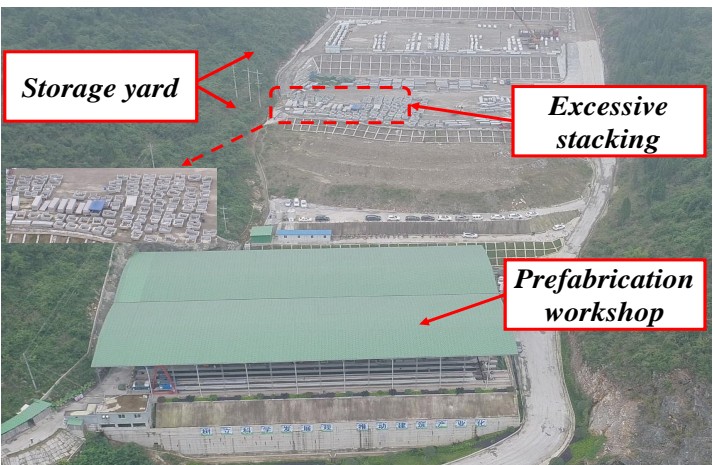

**Figure 2.** Aerial map of prefabrication workshop and storage yard.

Furthermore, it is worth noting that a PCE needs to be scanned 4 times to ensure sufficient accuracy. When multiple PCEs are scanned simultaneously as shown in Figure 3, it will save half of the scans for every two PCEs. Suppose that there is a total of $k$ PCEs, $4k$ times of scans are required if each PCE is scanned individually. However, only $2k + 2$ scans are needed if all PCEs are scanned simultaneously, which can easily save $2k - 2$ scans. Obviously, the more PCEs are scanned, the more time is saved. Therefore, scanning the multiple PCEs simultaneously is certainly a trend of PCE inspection in the future. However, as the scan data contains multiple PCEs, the efficient and accurate segmentation and recognition of different types of PCEs are essential for the consequent quality inspection.

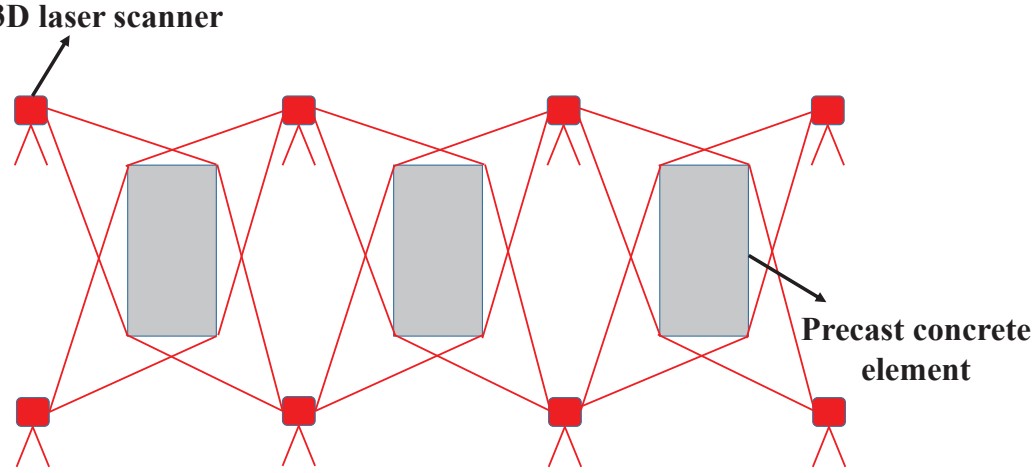

**Figure 3.** An illustrative example of multiple PCEs scanned simultaneously.

*2.2. Related Works*

2.2.1. Laser Scan Data Segmentation

Laser scan data segmentation aims to cluster points with similar characteristics into homogeneous regions, which could be helpful for analyzing the scanned objects in various aspects, such as position location and data recognition [21,22]. According to the perspective of segmentation functions, laser scan

data segmentation can be mainly divided into two types, which are surface segmentation and spatial segmentation, respectively.

For the surface segmentation, Bhanu et al. [23] first proposed to detect the edge of laser scan data by calculating the gradient information and detecting the change in the direction of the unit normal vector. Inspired by image segmentation, Besl and Jain [24] presented a 3D data segmentation method by using the idea of region growing algorithm. In addition, they achieved the segmentation by curvature calculation of each point and surface growing based on predefined criteria. In order to improve the robustness of the region-based method, Rabbani et al. [25] developed a robust laser scan data segmentation method, which combined the normal estimation with region growing methods. The control of the result from under-segmentation to over-segmentation was realized by adjusting curvature threshold and angle threshold. Che and Olsen [26] developed a segmentation method containing two steps, which were respectively the normal variation analysis and region growing. After detecting the noisy data by using normal variation analysis, region growing algorithm was used to grouped points on a smooth surface as a segmentation result. Maalek et al. [27] proposed a classification and segmentation technique for the laser scan data in construction site. A robust PCA method was first employed to extract planar and linear features for data classification. Then, the classified data was grouped by a robust complete linkage method. However, these surface segmentation methods require a large amount of computational burden, which are time-consuming and sensitive to noise. They cannot be used to isolate each scanned object, which are not suitable for PCE data segmentation.

Next, to achieve the spatial segmentation, the most popular methods are graph-based methods, which perform well on computing efficiency of data segmentation [28]. Golovinskiy and Funkhouser [29] extended the graph-cut segmentation by using k-nearest neighbors (KNN) to build a 3D graph. The edge weights were developed based on the exponential decay in length. By using the proposed segmentation method in [29], Golovinskiy et al. [30] developed a recognition method based on the shape and contextual features. A large-scale experiment on the urban data had been conducted. However, it cannot be applied to recognize the PCEs with the same type but different shapes. Klasing et al. [19] proposed a method, termed as radially bounded nearest neighbor graph (RBNN) algorithm, for the efficient segmentation of non-connected laser scan data. The RBNN algorithm determined the class of points by considering the class of other points within neighboring threshold. Strom et al. [31] proposed a segment union criterion according to the color and surface normals of the laser scan data to achieve the segmentation of both indoor and outdoor scenes. However, the graph-based methods rely on a system to establish topological relationships in laser scan data. They fail to segment the laser scan data when the scanned objects are connected by a continuous string of points, such as the ground data. Due to the huge amount of outdoor laser scan data for PCE inspection, building topological relationships will also become slow. Therefore, the graph-based methods may be not applicable to the outdoor laser scan data segmentation.

In addition, converting 3D data into 2D images can leverage the great success obtained in image processing for handling 3D data. In the literature, Awadallah et al. [32] converted 3D data into 2D images and used an active contour model to extract the edge of a sparse noisy data, which showed a better performance on the laser scan data with a high level of noise than previous system, but it failed to process the data in complex scenarios. Adam et al. [33] developed an approach to resolve the co-plane object segmentation by combining the 3D data structural information with 2D images. It should be noted that PCE segmentation in outdoor laser scan data, which is characterized by noise and a huge amount of data, denotes the task of separating each scanned PCE from the background data. Although the image-based approach suffered from the complexity of scanned scenarios, this problem can be handled by the combination with the graph-based method. To the best of our knowledge, there is no relevant literature to extract the PCEs in outdoor laser scan data by using image processing technology, such as clustering and edge detection. Therefore, this study attempts to improve the segmentation efficiency of outdoor laser scan data by combining image processing methods with the graph-based method.

2.2.2. Object recognition in the AEC industry

Extensive studies have been conducted to recognize the scanned objects from the laser scan data in the architecture, engineering, and construction (AEC) industry. For instance, Vosselman et al. [34] reviewed several recognition techniques and indicated that data segmentation played an important role in the recognition. Pu and Vosselman [35] adopted the planar surface growing algorithm [34] to segment the laser scan data and recognized each surface data according to the predefined feature constraints (e.g., size constraint and position constraint). These methods have a good performance in the recognition of the specific surfaces such as doors and roofs, but they rely on the human knowledge about the definition of the feature constraints.

To reduce human intervention and avoid complex semantic definition, some studies have proposed corresponding methods for simple structure elements. In particular, Luo and Wang [36] used mathematical morphology to extract projective circle parameters of the orthogonal data slices, thereby detecting the round cross-section column. In addition, using the coarse rasterization and Hough Transform, a method for detection of columns in as-built buildings from laser scan data was developed by Díaz-Vilariño et al. [37]. However, these methods are only available for the simple and regular elements.

To recognize complicated structure components in the AEC industry, it is straightforward to consider the as-designed models in BIM as a tool for object recognition. A technique presented by Bosché [13] was applied to recognize steel structures from laser scan data by using 3D CAD models. It was used to make a dimensional compliance control of steel elements. Nguyen et al. [38] compared the 3D CAD data with the scan data to make dimensional inspection of the piping system. Sharif et al. [18] presented a model-based finding approach to realize the recognition of some objects in cluttered construction laser scan data. To improve the efficiency of object recognition, Chen et al. [39] developed a principal axe descriptor (PAD) for construction-equipment classification from outdoor laser scan data. This descriptor used the CAD-generated data as training data, and tested the scanned data. It achieved a higher recall rate and lower computation time compared to other descriptors in reference [40]. Since the 3D coordinates and color information of the scanned object could be obtained by using a laser scanner with a built-in camera, the acquisition of color information provided another way to process the scanned PCE data. Wang et al. [41] developed a technique of automated position estimation of the rebars in reinforced PCEs. The authors recognized rebars based on the RGB values of the laser scan data and as-designed models of the rebar. However, only limited study has been conducted to recognize the multiple types of objects [39,40]. In addition, PAD method is not applicable for PCE recognition considering the following two aspects: (1) the existence of noisy data; and (2) the task of PCE recognition requires the identification of elements with the same type but different shapes. To the best of our knowledge, none of the previous work involves the recognition of multiple PCEs with different types in outdoor laser scan data, this may due to the arbitrariness of both the type and position of the scanned PCEs in the scanning environment.

## 3. The Proposed Segmentation and Recognition Approach

This section provides detailed steps of the proposed automatic recognition approach. The flow chart of this approach is shown in Figure 4. The proposed approach consists of two steps. The step of data segmentation is first explained in Section 3.1. Next, in Section 3.2, the recognition of different types of PCEs using as-designed model is illustrated in detail.

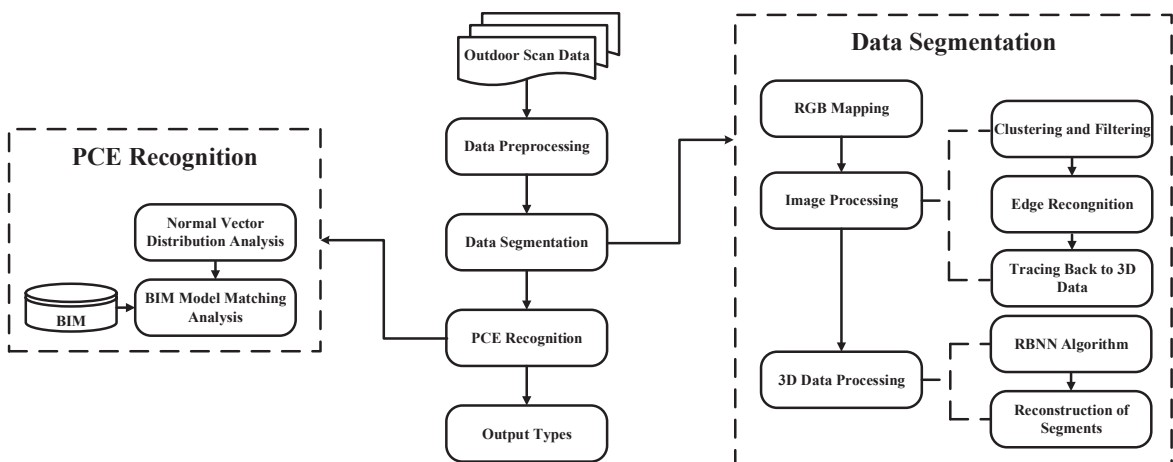

**Figure 4.** Flow chart of the proposed automatic recognition approach.

### 3.1. Data Segmentation

As shown in Figure 4, the data segmentation is carried out through three operations. First, a data preprocessing for ground data filtering is given in Section 3.1.1. Secondly, RGB mapping operation is explained in Section 3.1.2. Then, image processing methods used in this study is then illustrated in Section 3.1.3. Finally, in order to avoid under or over-segmentation, the judgement of the merge relationship for the reconstruction of segments is explained in Section 3.1.4.

### 3.1.1. Ground Data Filtering

It should be noted that there are huge number of ground points in the input data, because each scan contains a lot of ground laser scan data. Therefore, the RANSAC algorithm [42] is proposed to remove the ground data automatically. Since the large amount of data causes the huge computation burden, the entire input data is divided into several slices along the X and Y axes for RGB mapping. For example, when the data is mapped onto the $XZ$ plane, it can be sliced equidistant along the $Y$ axis.

### 3.1.2. RGB Mapping

To avoid traversal calculation of laser scan data in 3D space, 3D data is converted into 2D images based on RGB values of laser scan data. It is worth noting that the color data obtained by the TLS has different values in the r, g, b dimensions, but the grayscale data has the same value in these three dimensions. In other words, the grayscale data can be regarded as a special type of color data.

The row of the image is the $X$ or $Y$ axis direction in the laser scan data, and the column of the image is the $Z$ axis direction in the 3D data. Taking account of the mapping efficiency, the same number of grids $N$ is selected for both the row and column of the mapping image in the proposed approach. For instance, the following is to explain the use of $YZ$ plane to perform RGB mapping. The number of girds $N$ is an input parameter that needs to be determined initially. Then, the distance corresponding to the image grid can be obtained. In the horizontal direction and vertical direction, the gird distance $d_x$ and $d_z$ are calculated by Equations (1) and (2), respectively. In addition, the RGB average values of all points in a grid are considered to be the RGB values of the grid. When there is no point in the grid, the RGB values of this grid are replaced by 0. The image mapping effect and the running time with different $N$ are shown in Figure 5.

$$d_x = (x_{max} - x_{min})/N \tag{1}$$

$$d_z = (z_{max} - z_{min})/N \tag{2}$$

As the RGB mapping results shown in Figure 5, both higher and lower $N$ will increase running time and reduce image quality. When selecting a lower $N$, many points are gathered in a grid. It increases the running time of each grid for calculating the average RGB values. It indicates that

low image resolution will result in a poor image quality. On the other hand, a higher $N$ generates denser grids in the image, resulting in a slower point partition. Moreover, there will be more grids without laser scan data, which looks like spots in the image, further reducing the image quality. Hence, an appropriate $N$ is necessary for the RGB mapping. There are two criteria to select $N$: (1) Mapping images require enough pixels to ensure the image quality; (2) $N$ should not be too large to reduce the spots in the mapping images and the running time. The recommended value of $N$ in this study is within the range of 200~300 according to the mapping tests.

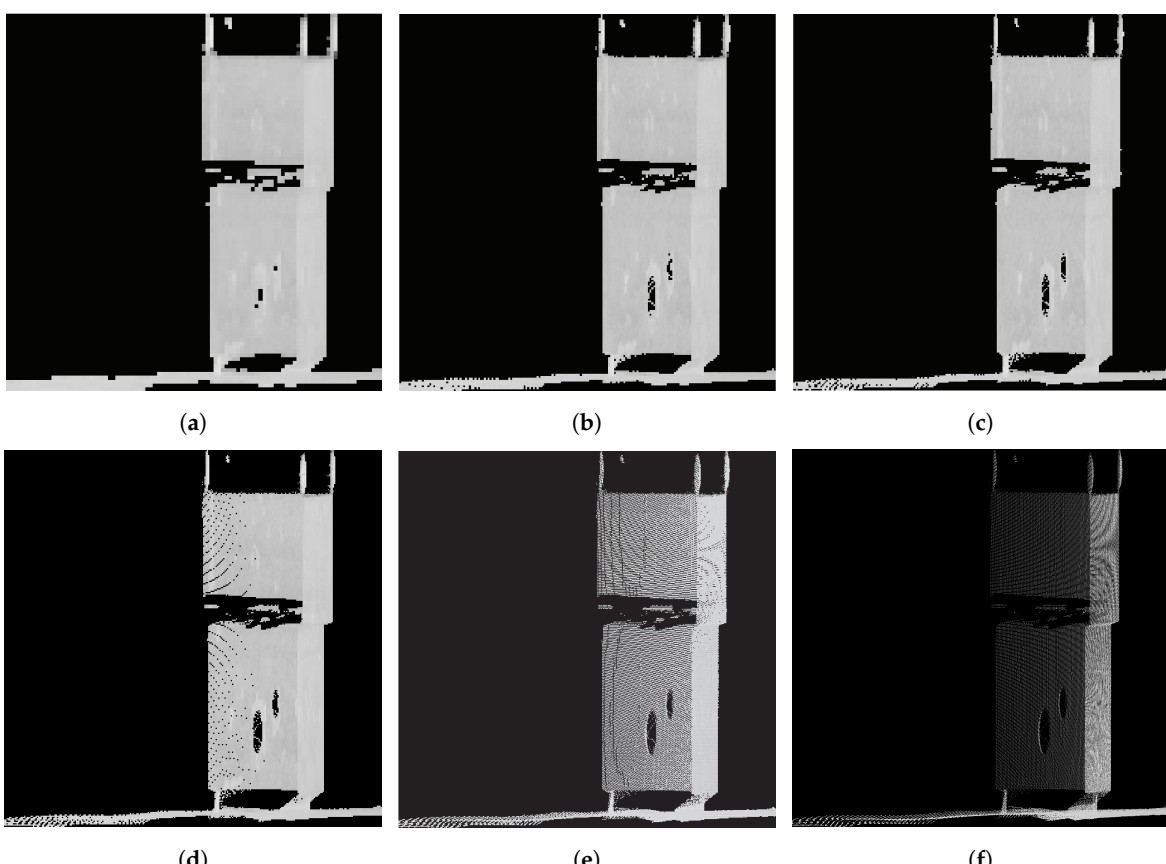

**Figure 5.** The results of RGB mapping tests with 268,221 points: (**a**) $N = 100$, $t = 0.84$ s. (**b**) $N = 200$, $t = 0.67$ s. (**c**) $N = 250$, $t = 0.66$ s. (**d**) $N = 300$, $t = 0.70$ s. (**e**) $N = 500$, $t = 0.86$ s. (**f**) $N = 1000$, $t = 1.24$ s.

### 3.1.3. Image Processing

This step is critical for a rapid laser scan data segmentation. Firstly, a morphological technique, known as opening and closing by reconstruction [43], is used to remove discrete pixel grids. In particular, morphological filtering only requires a small filter size in this study, because a large filter size will result in more data loss for low resolution mapping images. Then, an improved adaptive k-means algorithm [44] is used for clustering, which is based on the hierarchical clustering [45]. The use of clustering method aims to enhance the difference between the target data and background data. Then, edge recognition is conducted by using a canny operator [46] to transform the clustered image into the edge image. The purpose of using edge detection after clustering is to compensate for the fact that image segmentation is not necessarily accurate. Lastly, an active window algorithm is developed to extract the segmented data.

Clustering and Filtering

To facilitate image clustering, the image needs to be filtered at first. The morphological technique is simple and efficient for low resolution images. Therefore, the morphological technique with a small filter size (e.g., $3 \times 3$) is used in this study.

After the image is filtered, the k-means algorithm is employed to cluster the filtered image due to its high efficiency [45]. The key of the k-means algorithm is the selection of initial center and the formula of distance evaluation. Nevertheless, the k-means algorithm usually requires the number of clusters as an input parameter. To make the k-means algorithm more applicable, inspired by the hierarchical clustering, the formula for calculating the center point is modified. When a cluster is obtained, points in this cluster are removed from the total data according to the concept of hierarchical clustering. The algorithm loops until the stop condition is reached. The initial seed of a cluster can be determined by the highest frequency in the color histogram of the mapping image. Moreover, the influence of seed position and RGB values is considered when calculating the seeds. For ease of description, $(r, l, g)$ denotes the dimensions of the grid in the mapping RGB images, in which $r$, $l$, and $g$ denote the row, column, and RGB values of the grid, respectively. $(r_k, l_k, g_k)$ denotes the seed of the $k^{th}$ cluster, and each grid in the image can be calculated by Equation (3).

$$
\begin{aligned}
dist = \frac{1}{3}\{ &\sqrt{\alpha[(r_R - r_k)^2 + (l_R - l_k)^2] + \beta(g_R - g_k)^2} \\
+ &\sqrt{\alpha[(r_G - r_k)^2 + (l_G - l_k)^2] + \beta(g_G - g_k)^2} \\
+ &\sqrt{\alpha[(r_B - r_k)^2 + (l_B - l_k)^2] + \beta(g_B - g_k)^2}\}
\end{aligned}
\tag{3}
$$

where $\alpha + \beta = 1$, $\alpha$ and $\beta$ denote the position weight and the RGB values weight, respectively. It is obvious that appropriate edge information will be obtained by selecting suitable weights. Considering the low pixels of the actual mapping image in this approach, it is reasonable to consider a small position weight parameter, such as $\alpha = 0.01 \sim 0.1$.

Edge Recognition

As the difference between the background data and the PCE data is enhanced by clustering, the edge recognition technique, such as the canny operator, is employed to obtain edges between clusters and then get image segmentation results. As is known to all, the edge image is a logical image, and the grid of the edge is 1, otherwise it is 0. By locating edge grids in the edge image, the data within the grids is extracted in 3D space according to $d_x$ and $d_z$.

Tracing Back to 3D Data

In this study, to trace back to the laser scan data, we propose an active window algorithm to speed up the extraction of segmented data. Due to the morphological filtering, edges of the edge image will have some variations compared with that of the original image. Therefore, all edge grids for each object need to be included in a certain range for the 3D data extraction.

The main objective of the active window algorithm is to empty the edges to determine the range of each object, and the laser scan data in the traversal range is the segmented data. The active window is generated as a $2n + 1$ dimensional window, where $n$ is a positive integer. In the active window, the elements in the first and last row (column) are set as 1 and the rest of elements are set as 0. Taking an example of $n = 2$, the active window is shown in Figure 6.

$$\omega\big|_{n=2} = \begin{array}{|c|c|c|c|c|} \hline 1 & 1 & 1 & 1 & 1 \\ \hline 1 & 0 & 0 & 0 & 1 \\ \hline 1 & 0 & 0 & 0 & 1 \\ \hline 1 & 0 & 0 & 0 & 1 \\ \hline 1 & 1 & 1 & 1 & 1 \\ \hline \end{array}$$

**Figure 6.** Example of an active window with $n = 2$.

Before using this active window to process the edge image, the calculation rules need to be defined first, which are defined as follows:

- The central element in the active window should be placed at the calculated grid in the edge image. Other elements in the active window are placed at the corresponding grids in the edge image.
- When the size of surrounding elements of the calculated grid is less than the size of the active window, the surrounding elements will be supplemented by 0.
- All the selected elements in the first rule of the edge image are multiplied by the elements at the corresponding location in the active window, and the results replace the selected elements.
- The algorithm is stopped until all elements in the edge image become 0. A minimum rectangular range which can include all the tracks of the active window during the calculating process is taken as the output range of the elements in the edge image.

As shown in Figure 7, a 10 × 10 edge image processed by the active window (Figure 6) is set as an example of the application of these rules.

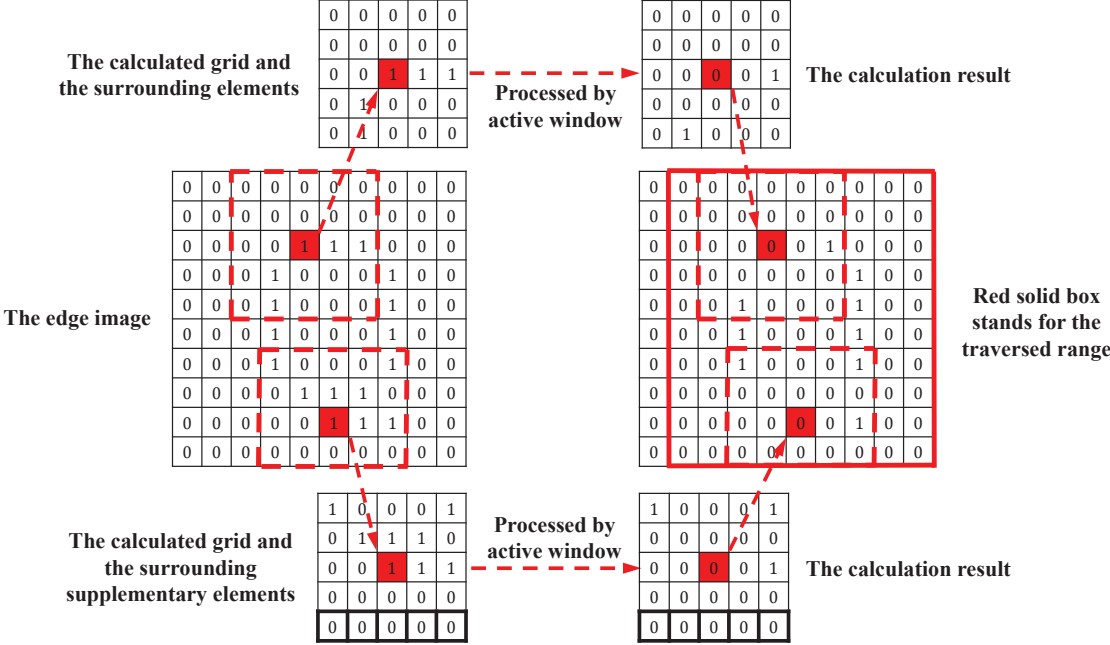

**Figure 7.** Example of using active window to process edge image.

In the edge image shown in Figure 7, two red-filled grids represent two typical calculated grids. The calculated grid in row 3 has complete surrounding elements, which are in the red dashed box, but the surrounding elements of the other one need to be supplemented by 0 as shown in black

solid box. In addition, the edge image on the right side shows the calculated results of two sample grids, and the red solid box is the output range of the edge image determined by the active window. According to the determined output range, points in this range will be extracted from the mapping image. Figure 8 gives an example of using the active window to extract each object in an edge image. The pseudo code of this method is given in Algorithm 1.

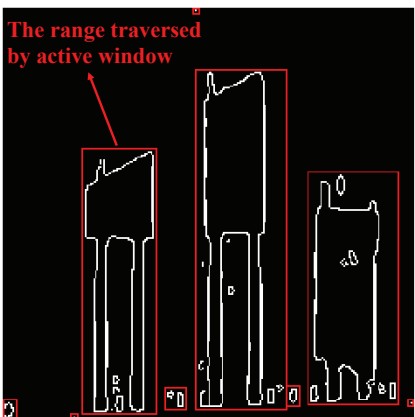

**Figure 8.** An example of extracting segments by the active window algorithm.

---

**Algorithm 1:** Active window algorithm

---

**Input:** Mapping image *I*, Edge image *E*, Active window size *n*
**Output:** Segmented data *D*
1:  $w$ = Create Active Window($n$);
2:  $[N, M]$ = Size($E$);
3:  **while** *E is NonZeros* **do**

4:      **for** $i = 1$ to $N$ **do**

5:          **for** $j = 1$ to $M$ **do**

6:              **if** $E(i,j)$ *is NonZeros* **then**

7:                  Initial Calculated Grid $indx$ = [i,j];
8:                  **repeat**

9:                      $W$ = Create Calculated Window($indx$);
10:                     **if** *Size of W is smaller than Size of w* **then**

11:                         $W$ = Supplementary Window($W$);
12:                     **end if**
13:                     $Result$ = Calculation($W, w$);
14:                     $E$ = The $Result$ is written into $E$;
15:                     $indx_r$ = NonZeros Index In $Result$($Result$);
16:                     $indx$ = NonZeros Index In$E$($Result$, $indx_r$)
17:                 **until** $indx$ *is Empty*
18:                 $R$ = The range traversed by active window;
19:                 $D$ = Create New Segmented Data($I$,$R$);
20:             **end if**
21:         **end for**
22:     **end for**
23: **end while**

---

### 3.1.4. 3D Data Processing

This section is the last step of automatic data segmentation. The goal of this section is to avoid under-segmentation or over-segmentation of the scanned PCE data. The RBNN algorithm is used to achieve a further segmentation, because it is highly efficient for the small amounts of data. The RBNN algorithm is controlled by a neighboring threshold and a small filtering threshold. In the case of

outdoor scanning environment, the neighboring threshold will no longer be 0.01 m as set in the reference [19]. A lager value is needed for calculational efficiency, such as 0.5∼1 m. In addition, the filtering threshold can be determined by the scanning density.

The flow chart of this merging calculation process is shown in Figure 9. To ensure the completeness of the PCE data, the merge judgement between every two segments is realized easily by setting two thresholds. $T_R$ is used to determine whether the two segments need to be judged, such as 5 m. $T_r$ is used to decide whether the two segments are adjacent, which is considered to be distance tolerance of adjacent segments, such as 5 cm. In particular, the centroid of each segment and the distance between the two centroids are calculated at first. Then, if the distance is less than $T_R$, continue to find whether there is a pair of points in the two segments whose distance is less than $T_r$. Two segments satisfying such conditions will be merged, and the centroid of the new segment will be further updated. In the end, the judgement is repeated until the centroid distance of all segments exceeds $T_R$. Therefore, the reconstruction of all segments is achieved.

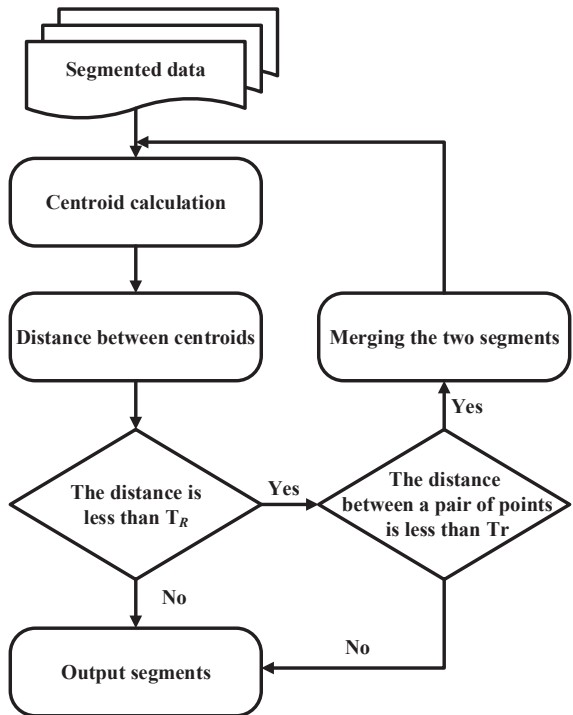

**Figure 9.** Flow chart of the merging calculation.

*3.2. Precast Concrete Element Recognition*

To recognize the types of the scanned PCEs, there is a premise that the PCE data must be as complete as possible, because the type of complicated PCEs (e.g., PC caisson toilet and PC exterior wall panel) cannot be accurately determined based on the obtained surface data. There is no doubt that the more complete the data is, the more accurate the data recognition is. Hence, in this study, the recognition of PCE data is conducted in registered outdoor laser scan data. To show the real inspection environment of PCEs, experiments are conducted in two different scanning environments. The first scanning environments in the prefabrication workshop with a lot of background, while the second one is a spacious environment in the storage yard, which has less background under a controlled manner. Comparing the recognition of PCE data in these two scanning environments, the main difference is that there is a large amount of non-PCE data in the first kind of environment. In Section 3.2.1, we illustrate the method of normal vector distribution analysis, which is employed to select the segment that may be the PCE data. The BIM model matching analysis is then given in Section 3.2.2

3.2.1. Normal Vector Distribution Analysis

Since most of the surfaces of PCEs are flat, the flat surfaces can be used as a typical feature for recognition. In order to filter the background segments, the normal vector distribution analysis is proposed. Considering the completeness of the multiple PCE data, the unit normal vectors of the external surface in PCE data will be concentrated in at least two directions. A high density of the unit normal vectors indicates that there is a plane in this vector direction. Therefore, there are generally two kinds of background data to be considered here. One is the scan data without any plane, which is characterized by the discrete normal vector distribution. The other one only has one flat surface, which has the normal vector distribution in a single direction centralization.

In this study, to illustrate the idea of normal vector distribution analysis, a PC exterior wall panel is taken as an example for explanation. The normal vector distribution of the PC exterior wall panel is compared with those of the two kinds of background data. Table 2 shows the parameters that need to be calculated for the normal vector distribution analysis and the comparison of the data in these three cases.

Figure 10 shows the implementation flow of the normal vector distribution analysis. First, unit normal vectors of the input data are calculated by PCA algorithm [47]. These unit normal vectors are represented by the Gauss map. As the length of each unit normal vector is 1, the vectors are transformed from the Cartesian coordinate system into the spherical coordinate system for dimensionality reduction. The unit normal vectors are represented by $\theta$ and $\varphi$, where $\varphi$ denotes the angle between the normal vector and the *XOY* plane, and $\theta$ denotes the angle between the projection of the unit normal vector on the *XOY* plane and the *X* axis. It is impossible to directly judge the PCEs by the discrete scatter plot of this normal vector distribution. A quantitative indicator for PCE judgement is thus required. Next, by converting the scatter plot into a $400 \times 400$ normal vector distribution image, it is observed that there are at least two bright areas of the PCE data where color values exceed 60 of the color bars. Furthermore, the normal vector distribution image is represented as a three-dimensional histogram. A frequency ratio $\eta$ of each grid is defined to reflect the density of the points with the approximate normal direction in the histogram as calculated by Equation (4).

$$\eta = N_{frequency} / N_{total} \times 100\% \tag{4}$$

where $N_{frequency}$ denotes the frequency in each grid of the histogram, and $N_{total}$ denotes the number of total data in the analyzed input data. Finally, the area where $\eta$ exceeds 1% is regarded as the bright area, and the number of the bright areas are counted by a clustering method. It is clear that the data with at least two bright areas will be output.

Figure 11 shows the examples of two kinds of background data with their normal vector distribution histograms. Figure 11a represents the data with the discrete normal vector distribution. It is worth mentioning that its maximum frequency is much smaller than that of PCE data. Moreover, Figure 11b represents the data with the normal vector distribution in a single direction centralization. The specific analysis information about these three examples is shown in Table 2 with respect to the number of bright areas $N_{areas}$, the peak frequency of an bright area $N_{local}$ and the peak frequency ratio of the bright area $\eta_{local}$. $N_{areas}$ is used as a basis for the initial selection of PCE data, which reduces the computation for PCE type identification.

**Table 2.** Analysis information for sample laser scan data.

| Laser Scan Data | $N_{total}$ | $N_{local}$ | $\eta_{local}(\%)$ | $N_{areas}$ |
|---|---|---|---|---|
| The data in Figure 10 | 281,270 | 14,542<br>4005 | 5.17<br>1.42 | 2 |
| The data in Figure 11a | 99,747 | 76 | 0.08 | 0 |
| The data in Figure 11b | 184,866 | 29,055 | 15.72 | 1 |

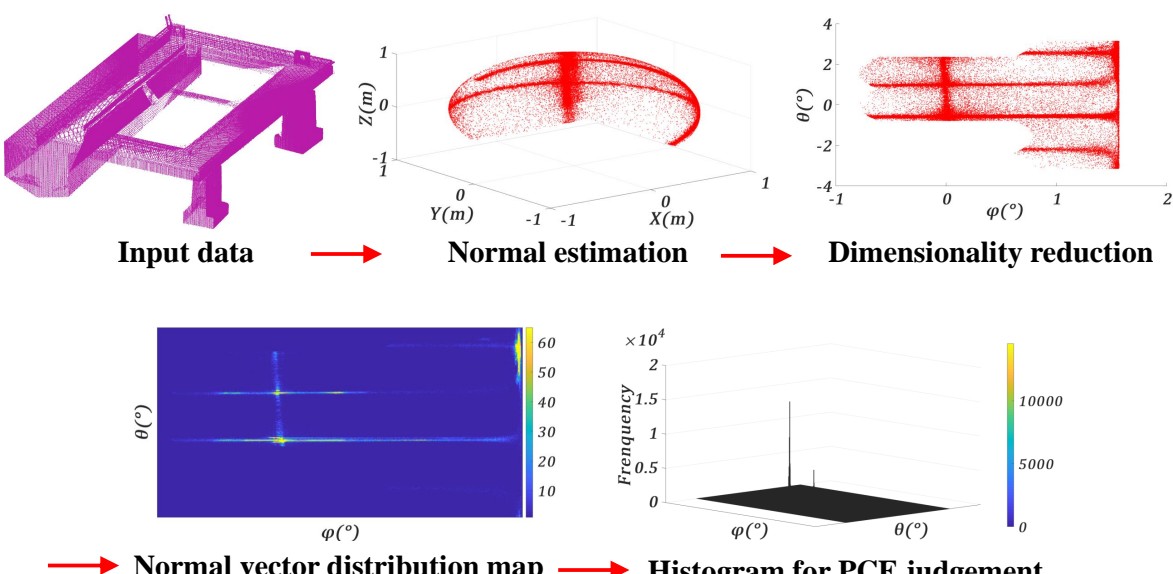

**Figure 10.** Each step for the normal distribution analysis.

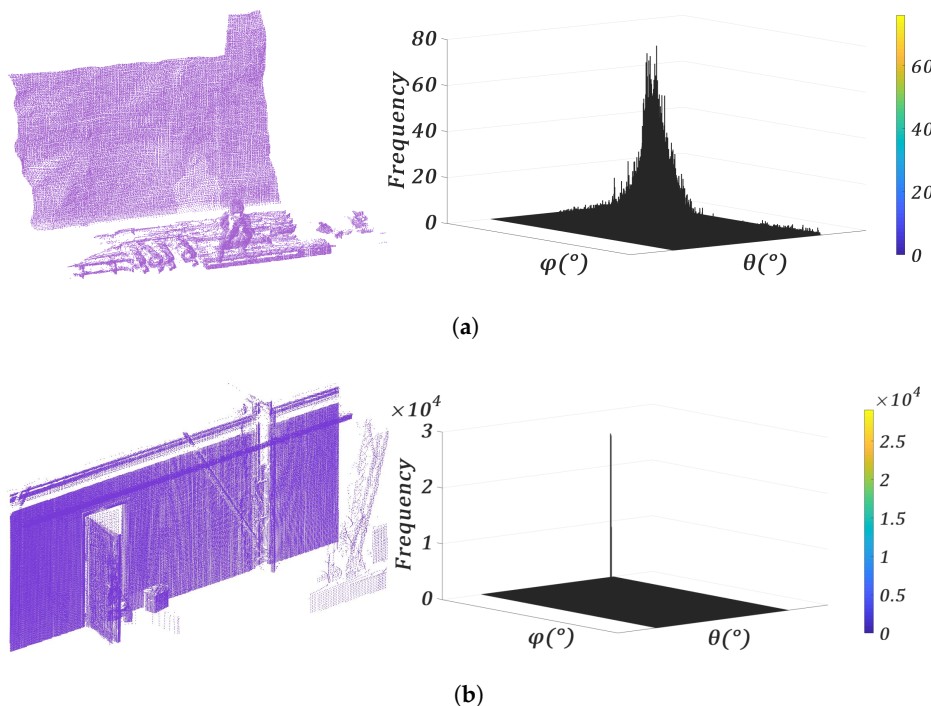

**Figure 11.** Examples of the two kinds of background data: (**a**) The background data with discrete normal vector distribution; (**b**) The background data with the normal vector distribution in a single direction centralization.

### 3.2.2. BIM model Matching Analysis

During the scanning of the multiple PCEs, the type of each PCE is known. In this study, these types are regarded as the prior knowledge to find corresponding as-designed models for further recognition of PCE data. A PC exterior wall panel shown in Figure 12a is taken as an example to illustrate the detailed BIM model matching analysis. As the recognition approach is based on the as-designed model, there is still an issue with the type selection of the model data. As is known to all, there may be some unscannable surfaces during the scanning, such as the bottom or inner surface of PCEs.

In addition, as shown in Figure 12b, there may also be some support data that is connected to the PCE data, because they cannot be thoroughly split. Hence, the proposed recognition approach suggests using the contour data of the as-designed model instead of the surface data to match with the PCE data. The contour data can reduce the influence caused by the missed or occluded surface data. Therefore, the contours of the as-designed model are extracted and discretized into points, as shown in Figure 12c,d.

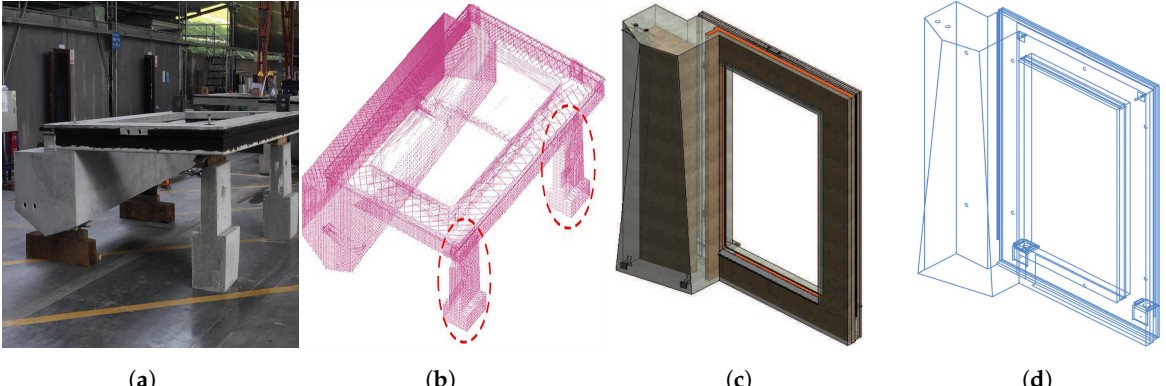

|             |             |             |             |
|:-----------:|:-----------:|:-----------:|:-----------:|
| (**a**)     | (**b**)     | (**c**)     | (**d**)     |

**Figure 12.** The scanned PC exterior wall panel and the as-designed model: (**a**) The scanned PC exterior wall panel in prefabricated plant; (**b**) The scanned PC exterior wall panel with support data; (**c**) The as-designed model in BIM; (**d**) The contour data of the as-designed model.

Each segment obtained by initial selection in Section 3.2.1 will be matched with the contour points. A coarse matching is achieved by the 4-points congruent sets (4PCS) algorithm [48]. In order to reduce time consumption, a sampling method known as max leverage sampling [49] is used to sample the segment and the contour points before the coarse matching. The 10% of the PC exterior wall panel data and the contour points sampled by this method is shown in Figure 13, in which the sampled data in blue is superior to the original data to emphasize the characteristics of the method. The max leverage sampling method is employed to extract the prominent features in the segment based on the leverage values sorted from large to small. In addition, by setting the minimum distance of the four randomly selected matching points (e.g., more than 0.5 m), the success rate of the coarse matching can be improved. Based on the concept of RANSAC algorithm, the matching result with higher coarse matching accuracy will be automatically output. Obviously, the contour data cannot be matched with the segment in different shape due to their different features. Therefore, when the multiple scanned PCE data have significant shape differences, the PCE types can be identified directly by using max leverage sampling method and 4PCS method. However, if the shape differences between the scanned PCEs are small, it is necessary to use ICP algorithm [13,50] for fine matching before further evaluation. Figure 14 shows the results of coarse matching and fine matching for the PC exterior wall panel.

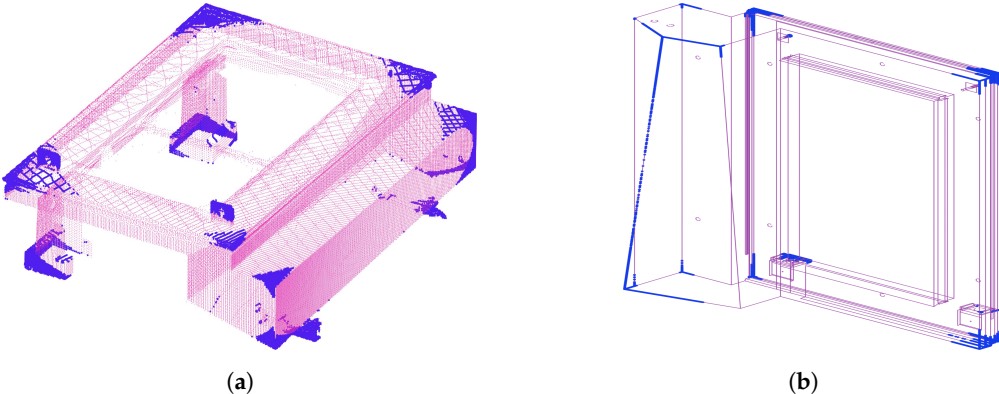

|                 |                 |
|:---------------:|:---------------:|
| (**a**)         | (**b**)         |

**Figure 13.** Examples of max leverage sampling shown as the blue data points: (**a**) The scanned PC exterior wall panel; (**b**) The contour points of the as-designed model.

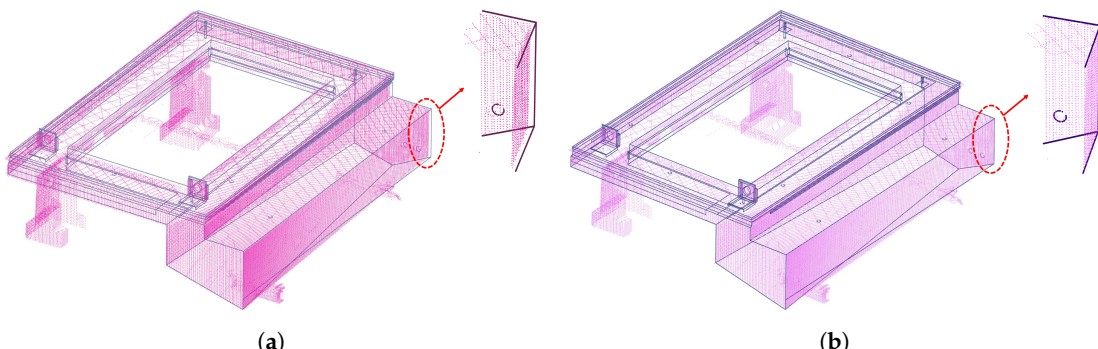

(**a**)          (**b**)

**Figure 14.** Matching results of the PC exterior wall panel data and the contour points of the as-designed model: (**a**) Coarse matching by using max leverage sampling method and 4PCS method; (**b**) Precise matching by ICP algorithm.

To further identify PCEs with small shape differences, based on the fine matching results, the degree of completeness (*DOC*) proposed by Wang et al. [41] is adopted in this study as the evaluation. The *DOC* value can be calculated by Equation (5).

$$DOC = A_{match} / A_{total} \tag{5}$$

where $A_{match}$ denotes the number of matched contour points and $A_{total}$ denotes the number of the total contour points. When calculating the $A_{match}$, the distance between each point in the contour points and the nearest point in the matched PCE data is calculated at first. Then, the number of all points whose distance is less than a preset threshold $d_{rad}$ (e.g., 2 cm) is counted. It goes without saying that the higher *DOC* value is, the better matching between the scanned data and the matching model is. Hence, the automatic recognition of multiple PCEs in outdoor laser scan data and type identification are realized based on the value of *DOC*.

## 4. Experiments on the Scanning of Multiple PCEs Simultaneously

To verify the effectiveness of the proposed automatic segmentation and recognition approach, experiments were conducted in terms of three aspects. In total, 22 PCEs were scanned, including 3 PCEs with different shapes, 3 PCEs with different types, and 16 PCEs (including 9 types) which were scanned simultaneously to verify the efficiency of proposed method in the case of a large number of PCEs. For the previous two aspects, PCEs were scanned in the prefabricated plant, which was the first scanning environment. They were all scanned in both color and grayscale. The same time was spent in scanning the color and grayscale data, while the grayscale scanning provided a higher scan density. In addition, for the last aspect, PCEs were scanned in the storage yard, which was regarded as the second scanning environment. Section 4.1 gives the details of these experiments. The outdoor laser scan data segmentation results are then shown in Section 4.2. The recognition results of different types of PCEs are given in Section 4.3.

### 4.1. Validation Experiment

Experimental Data Information

There were 5 outdoor laser scan data that were tested by our proposed approach. The designation of experimental data was assigned according to the classification of the color information and the experimental aspects. For example, $GSD - 1$ represents the grayscale scan data in terms of the first aspect, and $CSD - 1$ is the color scan data in terms of the first aspect.

These PCEs were scanned by a TLS, and all scanned PCEs kept a certain distance from each other during the scanning process. The PCEs were placed on the precast sleepers with a total height of 0.3 m. The specific types of these scanned PCEs are shown in Table 3. The floor and type of the PC exterior wall can be respectively recognized by the first and last number of its name. If the types of PC exterior wall panels are different, the main dimensions and shapes are different. In addition, panels with the type on different floors may have different shapes or dimensions. Only red numbered PC exterior wall panels with the same type in Table 3 have the same dimensions. Figure 15 shows the different dimensions of these PC exterior wall panels. In addition, experimental photos of these three aspects are shown in Figure 16. Figure 16a presents 3 PC exterior wall panels are scanned, in which two panels are identical and the main difference from the third one is the embedded parts. Meanwhile, 3 different types of PCEs are scanned as shown in Figure 16b. At last, a large number of PCEs are scanned simultaneously, such as PC stair, beam, column and many different PC exterior wall panels as shown in Figure 16c.

**Table 3.** The specific type information of five experiments.

| Experimental Data | Number of PCEs | PCE Type | Type Number | |
|---|---|---|---|---|
| $GSD-1$ $CSD-1$ | 3 | Panel | $9W03-1$ $9W03-2$ | |
| $GSD-2$ $CSD-2$ | 3 | Panel Stair Cassion toilet | $9W03-1$ $TB3$ $WS02$ | |
| $GSD-3$ | 16 | Beam Column Stair Cassion toilet Slab | $R9-2S11A$ $R3C03$ $TB2$ $WS01X$ $B4$ | |
| | | Panel | T-08 | 7W08 10W08 11W08 12W08 13W08 |
| | | | T-07 | 3W07 9W07 10W07 11W07 |
| | | | T-06 | 4W06 |
| | | | T-02 | 10W02 |

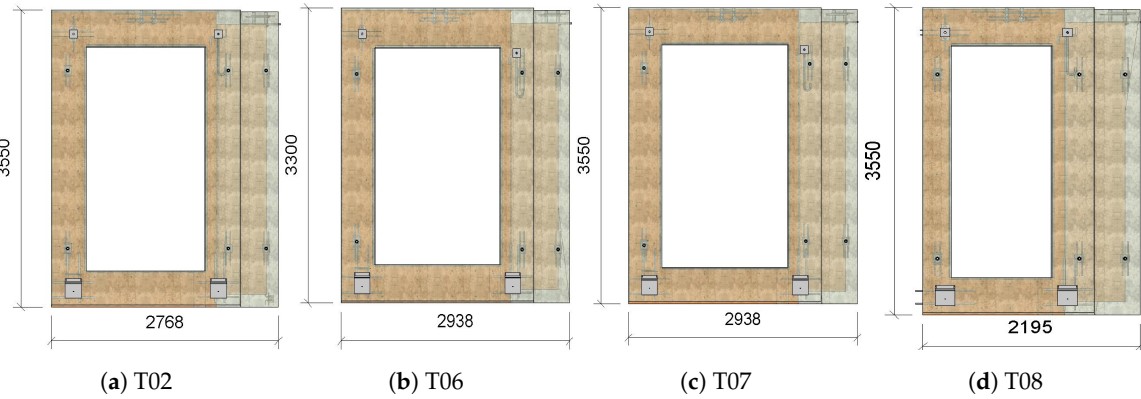

| 3550 | 3300 | 3550 | 3550 |
|---|---|---|---|
| 2768 | 2938 | 2938 | 2195 |
| (**a**) T02 | (**b**) T06 | (**c**) T07 | (**d**) T08 |

**Figure 15.** Main dimensions of different types of scanned PCEs.

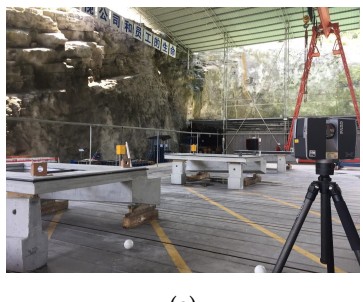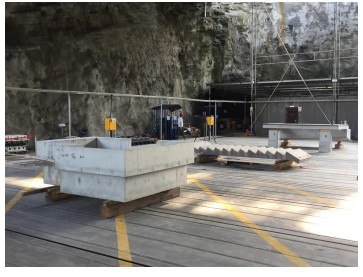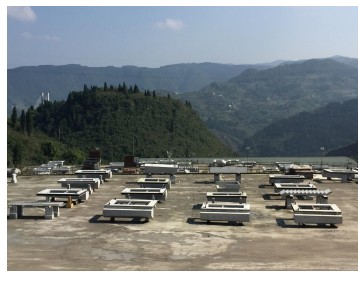

| (**a**) | (**b**) | (**c**) |

**Figure 16.** Experimental photos of the three aspects: (**a**) PCE type; (**b**) PCE shape; (**c**) PCE amount.

*4.2. Segmentation Results*

In these experiments, the proposed approach was compared with the RBNN algorithm. Both methods used the same neighboring threshold (1 m) and filtering threshold (1000). Since the RBNN algorithm could not remove the ground data automatically, the ground data in each experimental data was first removed by the RANSAC algorithm. Table 4 summarizes the information of these five experimental data, where $N_{total}$ denotes total number of points and $N_{NG}$ represents the number of non-ground points. Figure 17 shows the segmentation results, and each segment is shown in different colors. Moreover, as shown in Figure 18, all the segments obtained by the proposed approach were numbered to facilitate the analysis of the segmentation results.

**Table 4.** The specific type information of five experiments.

| Experimental Data | $N_{total}$ | $N_{NG}$ |
|:---:|:---:|:---:|
| $GSD-1$ | 10,111,120 | 3,164,312 |
| $CSD-1$ | 3,842,303 | 1,195,405 |
| $GSD-2$ | 11,340,361 | 3,115,780 |
| $CSD-2$ | 4,370,985 | 1,170,248 |
| $GSD-3$ | 141,342,932 | 28,571,678 |

The comparison of the two methods is given in respect of running time and the number of segments as shown in Figure 19. In order to evaluate the difference between the segmentation results of the two methods, a comparison method on each segmented PCE data is proposed for error evaluation according to reference [51]. First, the ground truth of each PCE data in these experimental data is extracted manually, and the number of the extracted data is denoted as $N_{GT}$. Then, the number of error points for each segmented PCE data is defined as the sum of false positive ($FP$) and false negative ($FN$). Next, the error rate of each PCE data $Er$ is calculated by Equation (6).

$$Er = (FP + FN)/N_{GT} \tag{6}$$

Finally, in each experimental data, these $Er$s are normalized based on the maximum error rate which is calculated by the proposed approach and the RBNN algorithm for all segmented PCE data. The error evaluation results are shown in Figure 20, in which the numbers on the $X$ axis represent the number of PCE data shown in Figure 18.

Table 4 shows that these non-ground data accounts for $20 \sim 30\%$ of the total data. With the same scanning time, the amount of grayscale scan data is about triple of the amount of color scan data. Figure 17 shows that the proposed approach in this paper can realize the segmentation of outdoor laser scan data including multiple PCEs. It is indicated that both methods can segment laser scan data in color or grayscale. Therefore, when the scanning time is the same, it is recommended to scan multiple PCEs simultaneously in grayscale due to the higher scan density.

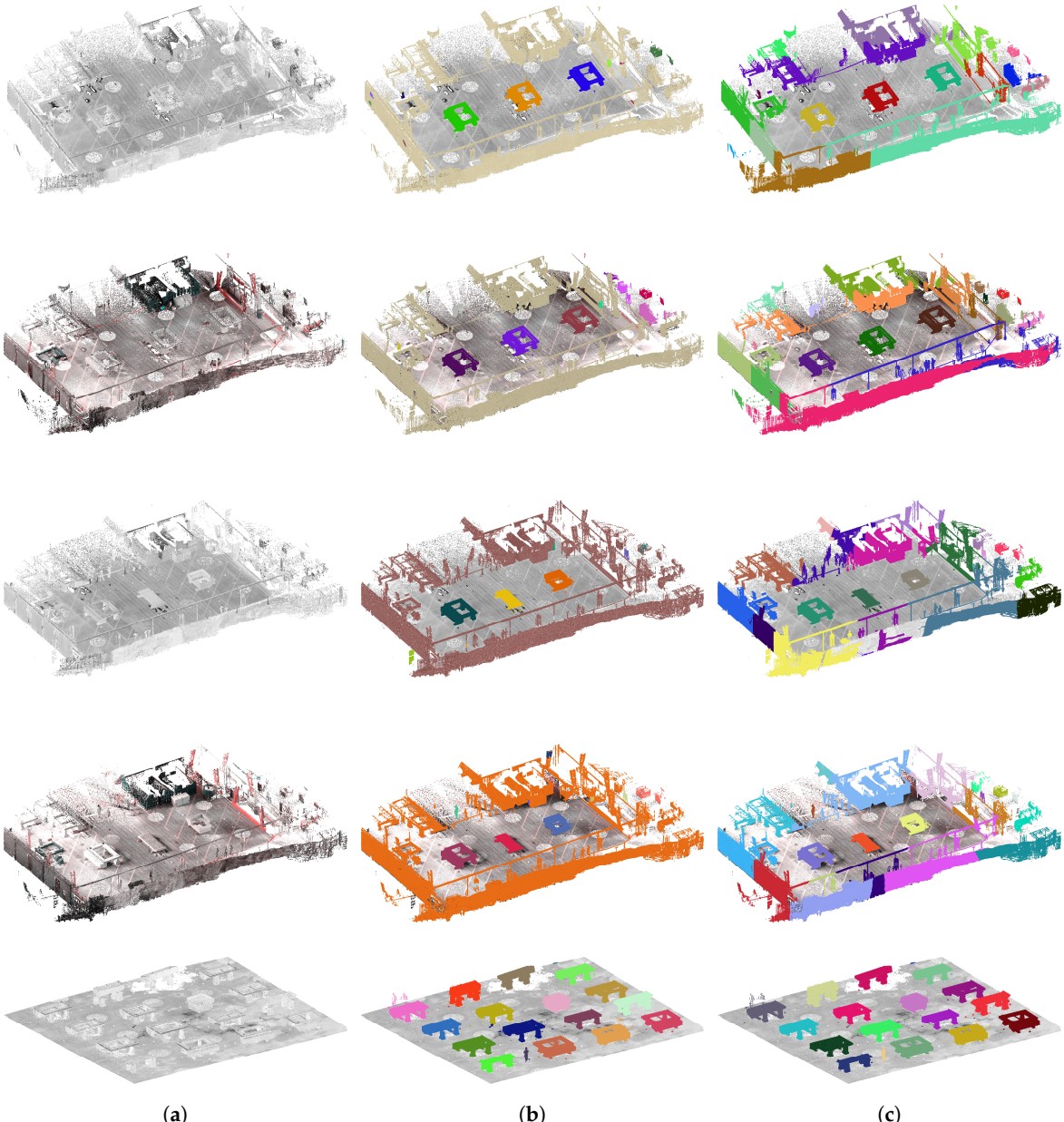

(**a**) (**b**) (**c**)

**Figure 17.** Segmentation results of PCEs in outdoor laser scan data. The first to fifth rows show the segmentation results in experimental data $GSD-1$, $CSD-1$, $GSD-2$, $CSD-2$, and $GSD-3$, respectively: (**a**) Original laser scan data; (**b**) Segmentation results processed by the RBNN algorithm; (**c**) Segmentation results using the proposed approach.

As shown in Figure 19a, the proposed approach in this study provides faster segmentation because the image processing technology reduces the data processing time. The time spent by the RBNN algorithm is about 3 times of that the spent by the proposed approach. It is demonstrated that the proposed approach improves the processing speed for large data volumes. However, the proposed approach fails to have good performance in the segmentation of background data, as shown in Figures 17a and 19b. Meanwhile, the RBNN algorithm has the under-segmented results for the background data. The numbers of segments obtained by the proposed approach are more than those obtained by the RBNN algorithm, considering the fact that the proposed approach does not consider the segments whose centroid distance are more than 5 m. Nonetheless, the background data is not the focus of this study. This issue can be solved by scanning in a controlled manner as in the second kind of scanning environment, as shown in the fifth row in Figure 17. Figure 21 shows the

manual segmentation result of $GSD - 3$ for PCE data. There are 17 segments selected manually in this experimental data. Due to the existence of filtering operation, the segmentation result of the proposed approach is better in line with the manual segmentation result in contrast to the RBNN algorithm according to Figures 17 and 19b.

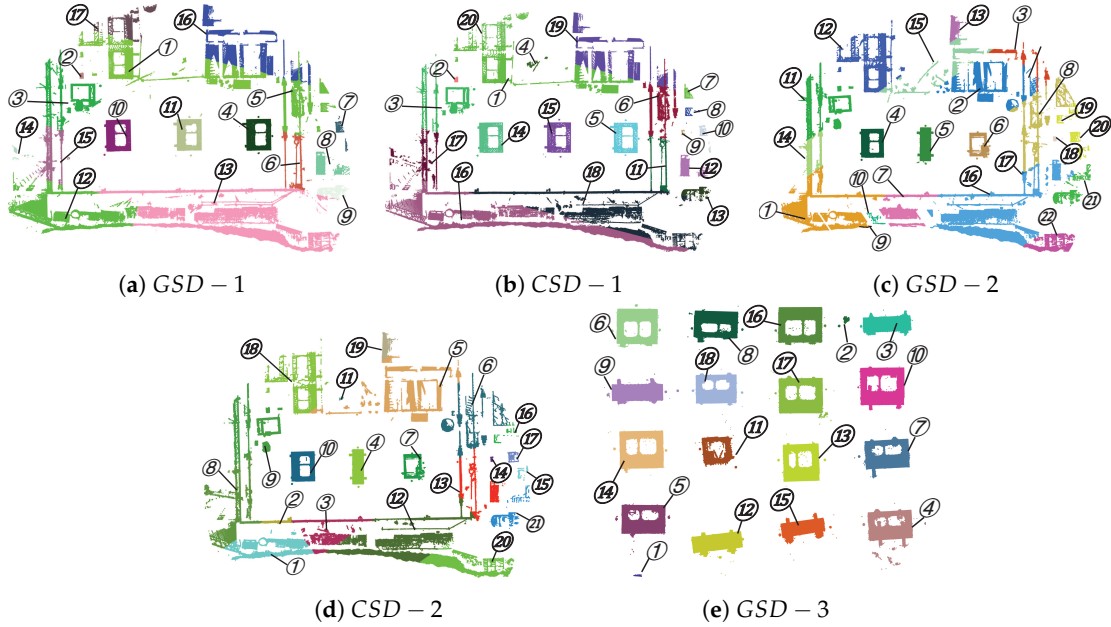

(**a**) $GSD - 1$　　　　　　　　　(**b**) $CSD - 1$　　　　　　　　　(**c**) $GSD - 2$

(**d**) $CSD - 2$　　　　　　　　　(**e**) $GSD - 3$

**Figure 18.** The number of each segment in segmentation results.

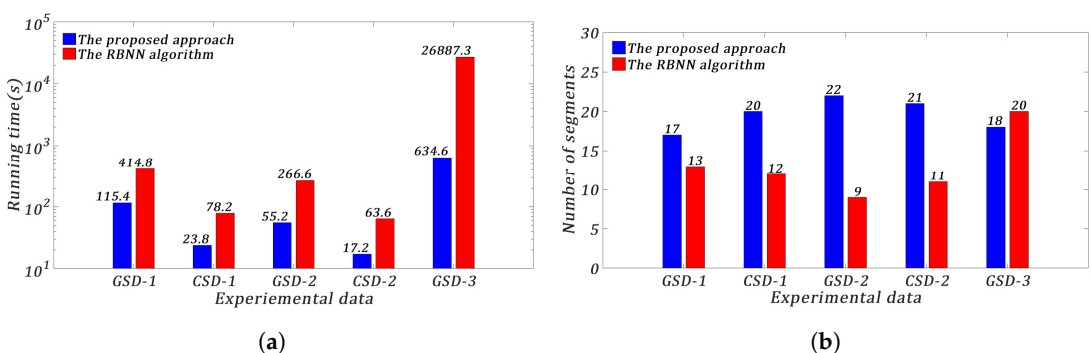

(**a**)　　　　　　　　　　　　　　　　　　　(**b**)

**Figure 19.** Comparison of the proposed approach and the RBNN algorithm: (**a**) Running times of each experimental data; (**b**) Number of segments of each experimental data.

Figure 20 presents the comparison of normalized segmentation errors of the two methods on the PCE data in each experimental data. In these histograms, 1 represents the result with the largest error. Figure 20a,b show the error evaluation of the experiment with respect to PCE shape and PCE type, respectively. The two methods have similar segmentation errors for most of PCE data, but the maximum error rate usually belongs to the result of the RBNN algorithm. For the experiment of PCE amount, $Er$ of the PCE data obtained by the RBNN algorithm is generally higher than that obtained by the proposed approach, because it is impossible to effectively remove noisy data in the outdoor scenes. The proposed approach has a better performance in the segmentation of the PCE data in the outdoor laser scan data than the RBNN algorithm, this is because a filtering step in image processing is adopted in the proposed approach.

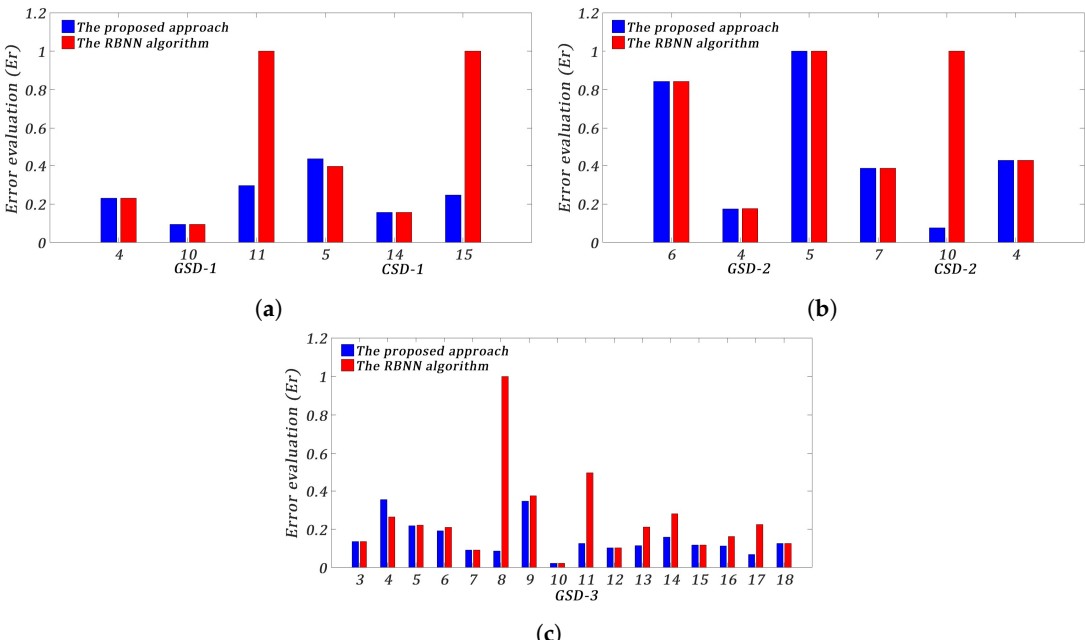

**Figure 20.** Comparison of error evaluation of PCE data obtained by the proposed approach and the RBNN algorithm: (**a**) PCE shape; (**b**) PCE type; (**c**) PCE amount.

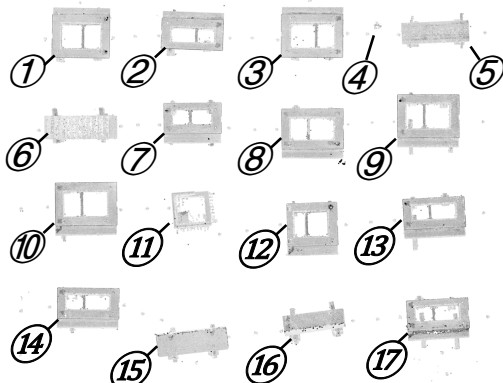

**Figure 21.** The manual segmentation result of the $GSD - 3$.

*4.3. Recognition Results*

Considering the previous four experimental data scanned in the first scanning environment, normal vector distribution analysis is conducted on these four experimental data. The calculation result of $N_{areas}$ of each segment in these four experimental data is shown in Figure 22. As aforementioned, the segmented data with $N_{areas}$ greater than 1 is considered to be PCE data. Therefore, the red line in Figure 22 is regarded as the selection limit of the normal vector distribution analysis results, and the value of $N_{areas}$ exceeding the red line is selected.

According to the analysis results, the segments which are potential PCE data are selected, i.e., 8 segments in $GSD - 1$ (No. 1, 2, 4, 7, 10, 11, 14 and 16); 6 segments in $CSD - 1$ (No. 1, 2, 5, 9, 14 and 15); 7 segments in $GSD - 2$ (No. 3, 4, 5, 6, 12, 18 and 20); 4 segments in $CSD - 2$ (No. 4, 7, 10 and 18). The normal vector distribution analysis has good performance in quickly distinguish the probable PCE data from background data, but the PCE types cannot be accurately identified. Therefore, it needs to use the as-designed model for coarse matching and fine matching to further identify the selected segments.

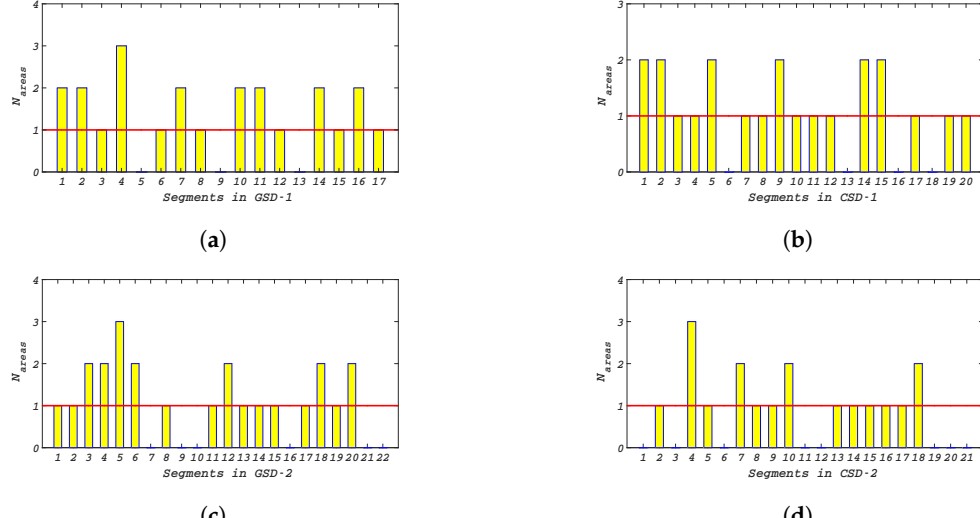

**Figure 22.** Normal vector distribution results: (**a**) 8 segments selected in $GSD - 1$. (**b**) 6 segments selected in $CSD - 1$. (**c**) 7 segments selected in $GSD - 2$. (**d**) 4 segments selected in $CSD - 2$.

The corresponding as-designed models used for automatic recognition are discretized into points, and the number of contour points are shown in Tables 5–8. After coarse and fine matching between all the selected segments and the contour points of the as-designed model, the *DOC* values of all successfully matched segments are given in Tables 5–8. In particular, in order to facilitate the presentation of the recognition results in the experiment of PCE amount, Table 7 shows segments which are successfully matched with multiple as-designed models, and Table 8 shows segments which are only matched with only one as-designed model. In addition, segments that cannot be coarsely matched to any contour data will have no DOC value. Hence, based on the *DOC* values, the prediction for each PCE data is presented.

**Table 5.** The evaluation value *DOC* of each selected segment in the experiment of PCE shape.

| Experimental Data | | $GSD - 1$ | | | | | | | | $CSD - 1$ | | | | | |
|---|---|---|---|---|---|---|---|---|---|---|---|---|---|---|---|
| Number | | 1 | 2 | 4 | 7 | 10 | 11 | 14 | 16 | 1 | 2 | 5 | 9 | 14 | 15 |
| $N_{total}$ | | 534,529 | 4878 | 99,810 | 7029 | 114,124 | 103,919 | 2252 | 173,061 | 203,429 | 1859 | 38,089 | 1770 | 43,700 | 39738 |
| $9W03 - 1$ | $A_{match}$ | - | - | 45,928 | - | 46,555 | 46,143 | - | - | - | - | 41,092 | - | 42,907 | 42,600 |
| $(A_{total} = 53,886)$ | $DOC(\%)$ | - | - | 85.23 | - | 86.40 | 85.63 | - | - | - | - | 76.26 | - | 79.63 | 79.06 |
| $9W03 - 2$ | $A_{match}$ | - | - | 46,010 | - | 46,436 | 46,087 | - | - | - | - | 41,166 | - | 42,157 | 40,163 |
| $(A_{total} = 53,834)$ | $DOC(\%)$ | - | - | 85.47 | - | 86.26 | 85.61 | - | - | - | - | 76.47 | - | 78.31 | 74.61 |
| Prediction | | - | - | 2 | - | 1 | 1 | - | - | - | - | 2 | - | 1 | 1 |
| Actual type | | - | - | 2 | - | 1 | 1 | - | - | - | - | 2 | - | 1 | 1 |

**Table 6.** The evaluation value *DOC* of each selected segment in the experiment of PCE type.

| Experimental Data | $GSD - 2$ | | | | | | | $CSD - 2$ | | | |
|---|---|---|---|---|---|---|---|---|---|---|---|
| Number | 3 | 4 | 5 | 6 | 12 | 18 | 20 | 4 | 7 | 10 | 18 |
| $N_{total}$ | 24,455 | 301,777 | 119,004 | 132,635 | 87,637 | 3565 | 7182 | 45,946 | 50,595 | 115,313 | 31,444 |
| Matched model | - | $9W03 - 1$ | $TB3$ | $WS02$ | - | - | - | $TB3$ | $WS02$ | $9W03 - 1$ | - |
| $A_{total}$ | - | 53,886 | 33,409 | 46,587 | - | - | - | 33,409 | 46,587 | 53,886 | - |
| $A_{match}$ | - | 47,728 | 32,265 | 29,907 | - | - | - | 31,405 | 24,044 | 44,986 | - |
| $DOC(\%)$ | - | 88.57 | 96.58 | 64.20 | - | - | - | 94.00 | 51.61 | 83.48 | - |

As shown in Tables 5–8, all PCE data is accurately recognized as their actual type. The *DOC* values of $GSD - 1$ and $GSD - 2$ are compared to those of $CSD - 1$ and $CSD - 2$ in Tables 5 and 6, it is indicated that *DOC* values of the high-density scan data are larger than those of low-density scan data. This stands that a high-density scan data provides a better type identification evaluation. Besides, Tables 6 and 8 show that the second experimental aspect is the easiest case for recognition. In other words, when the experimental data contains only different types of PCE, we can directly obtain the

accurate PCE types by coarse matching. Moreover, *DOC* values of the segments after successful fine matching with multiple models are shown in Tables 5 and 7. The total number of contour points for the same types of PCE is relatively close. Therefore, according to the experiment of PCE shape, such as No. 11 in $GSD-1$ and No. 5 in $CSD-1$, even if the shape difference is small between $9W03-1$ and $9W03-2$, the *DOC* values are still available for the identification of the PCE type successfully.

**Table 7.** The evaluation value *DOC* of the similar segment in the experiment of PCE amount.

| Experimental Data | | $GSD-3$ | | | | | | |
|---|---|---|---|---|---|---|---|---|
| Number | | 4 | 5 | 7 | 8 | 10 | 14 | 16 |
| $N_{total}$ | | 1,500,955 | 1,542,872 | 1,665,727 | 1,692,552 | 1,729,999 | 1,974,485 | 2,113,982 |
| 9W07 | $A_{match}$ | - | - | - | - | 51,597 | 53,824 | 51,663 |
| ($A_{total}=63,247$) | $DOC(\%)$ | - | - | - | - | 81.58 | 85.10 | 81.68 |
| 10W07 | $A_{match}$ | - | - | - | - | 50,175 | 49,673 | 52,283 |
| ($A_{total}=62,465$) | $DOC(\%)$ | - | - | - | - | 80.32 | 79.52 | 83.70 |
| 11W07 | $A_{match}$ | - | - | - | - | 50,101 | 47,662 | 48,314 |
| ($A_{total}=60,073$) | $DOC(\%)$ | - | - | - | - | 83.40 | 79.34 | 80.43 |
| 10W08 | $A_{match}$ | 45,007 | 45,480 | 47,273 | 45,721 | - | - | - |
| ($A_{total}=60,066$) | $DOC(\%)$ | 74.93 | 75.72 | 78.70 | 76.12 | - | - | - |
| 11W08 | $A_{match}$ | 46,790 | 47,379 | 46,002 | 48,744 | - | - | - |
| ($A_{total}=62,820$) | $DOC(\%)$ | 74.48 | 75.42 | 73.23 | 77.59 | - | - | - |
| 12W08 | $A_{match}$ | 46,458 | 50,897 | 47,167 | 47,502 | - | - | - |
| ($A_{total}=62,523$) | $DOC(\%)$ | 74.31 | 81.41 | 75.44 | 75.98 | - | - | - |
| 13W08 | $A_{match}$ | 46,708 | 45,547 | 44,976 | 44,411 | - | - | - |
| ($A_{total}=58,215$) | $DOC(\%)$ | 80.23 | 78.24 | 77.26 | 76.29 | - | - | - |
| Prediction | | 13W08 | 12W08 | 10W08 | 11W08 | 11W07 | 9W07 | 10W07 |
| Actual type | | 13W08 | 12W08 | 10W08 | 11W08 | 11W07 | 9W07 | 10W07 |

However, it should be noted that there are several well-matched segments with a low *DOC* value, such as No. 6 in $GSD-2$, No. 7 in $CSD-2$ and No. 11 in $GSD-3$. Although these PCE data has been correctly recognized, *DOC* values of these PCE data are smaller than those of other PCE data obtained in the same experimental data. The type of these PCE data is the PC caisson toilet, which has both inner and outer surfaces. Obviously, for this type of PCE, a low *DOC* value will be obtained when the internal surface data is missing. On the other hand, as shown in Table 8, there is a PC superimposed slab that has a *DOC* value of 99.98%, this is because the PC superimposed slab does not have any internal surfaces or holes that cannot be scanned. Therefore, the proposed approach is suitable for the PCE data with fewer unscannable surfaces.

**Table 8.** The evaluation value *DOC* of the unique segment in the experiment of PCE amount.

| Experimental Data | | | | $GSD-3$ | | | | | | | |
|---|---|---|---|---|---|---|---|---|---|---|---|
| Number | 1 | 2 | 3 | 6 | 9 | 11 | 12 | 13 | 15 | 17 | 18 |
| $N_{total}$ | 7804 | 8868 | 952,500 | 1,575,529 | 1,707,636 | 1,788,907 | 1,793,559 | 1,817,340 | 2,075,280 | 2,245,653 | 2,312,013 |
| Matched model | - | - | B4 | 4W06 | TB2 | WS01X | R9−2S11A | 3W07 | R3C03 | 10W02 | 7W08 |
| $A_{total}$ | - | - | 28,612 | 53,226 | 33,409 | 30,038 | 19,963 | 46,275 | 14,339 | 61,033 | 53,287 |
| $A_{match}$ | - | - | 28,607 | 46,365 | 28,317 | 22,743 | 18,773 | 43,703 | 11,587 | 51,711 | 43,854 |
| $DOC(\%)$ | - | - | 99.98 | 87.11 | 84.76 | 75.71 | 94.04 | 94.44 | 80.81 | 84.73 | 82.30 |

## 5. Conclusions

In this study, in order to improve the data acquisition efficiency in the process of PCE quality inspection, an automatic segmentation and recognition approach has been proposed for handling multiple PCEs in outdoor laser scan data. Based on the RBG values of laser scan data, the proposed approach is combined with the image processing technology and the RBNN algorithm to speed up the data segmentation. The coarse and fine matching between the PCE data and the as-designed model has been realized for the PCE type recognition. To validate the proposed approach, experiments on outdoor laser scan data have been conducted with respect to the shape, type, and amount of PCEs. For the evaluation on the PCE shape and type, the color and grayscale scanning with the same amount of time have been carried out.

From the experimental verification, we have the following conclusions: (1) In contrast to the RBNN algorithm, the proposed approach can save at least 2 times with respect to running time. The proposed approach can segment all the PCE data, but it has poor performance in the background data segmentation; (2) The proposed approach is available for both the grayscale and color scan data; (3) The normal vector distribution analysis significantly facilitates the recognition of PCE data in the first kind of scanning environment, which can reduce the number of matches that need to be made; (4) The proposed approach has significant effect on the PCE data with fewer unscannable surfaces.

However, the proposed approach will be influenced by the accuracy of precision matching algorithm and the integrity of the PCE data. In future work, the impact of these two aspects on the recognition results needs to be reduced. Further investigations are needed to extend its applicability to the simultaneous quality inspection of large batches of PCEs. We would like to use deep learning for an automatic segmentation and recognition of structural components in laser scan data of as-built buildings.

**Author Contributions:** Conceptualization, J.L., D.L. and L.F.; methodology, D.L. and L.F.; software, D.L. and P.L.; investigation, D.L. and W.W.; writing–original draft preparation, D.L.; writing–review and editing, L.F.; project administration, J.L.; funding acquisition, J.L.

**Funding:** This research was funded by the National Natural Science Foundation of China [grant numbers 51622802; grant numbers 61876025].

**Conflicts of Interest:** The authors declare no conflict of interest.

## Abbreviations

The following abbreviations are used in this manuscript:

| | |
|---|---|
| PCE | Precast concrete element |
| TLS | Terrestrial laser scanner |
| BIM | Building information modeling |
| RBNN | Radially bounded nearest neighbor graph |
| AEC | Architecture, engineering, and construction |
| RANSAC | Random sample consensus |
| PCA | Principal components analysis |
| DOC | Degree of completeness |

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
