# Peer review of "Towards Automatic Segmentation and Recognition of Multiple Precast Concrete Elements in Outdoor Laser Scan Data"

_remotesensing, doi:10.3390/rs11111383_

Round 1

Reviewer 1 Report

Dear authors,

I reviewed your paper titled (Towards Automatic Segmentation and Recognition of

Multiple Precast Concrete Elements in Outdoor Laser Scan Data) and I have the following comments:

Line 17:… use a coarse matching and a precise matching

Reviewer:may be coarse better replaced by and a fine matching

Line 19:… outdoor laser scan data

Reviewer:  since you mentioned a specific number of millions or ten million, then I prefer to say: laser scan points 

Line 22:  efficacy

Reviewer:  you mean efficiency!?

Line 22:  involved in this study

reviewer: you mentioned experimental studies so more than one study but then here you say this study!

I expect to write in these studies,

or you mean in this paper!

Line 54:  affected by the quality of the images and the lighting conditions

Reviewer:  still not convincing to me why the image based is not suitable as much as TLS!

with the current dense matching and reconstruction even with open source tools, we can achieve very reliable dense point clouds of the PCEs especially if we also considered the texture in the segmentation which is not available in the raw laser data.

Line 58:  0.3_0.4mm within 10m…

Reviewer: you need first to clarify the required dimensional accuracy of the PCEs according to international standards and then you discuss the accuracy that can be attained by TLS or images,

Line 99:  into 2D images, and perform..

Reviewer: here I am running into a conflict!! You mentioned that image based techniques are less sufficient compared with the TLS data, but now I am surprised that you are projecting the TLS data back to 2D images, so why not using images in the first place? You need to carefully clarify in the text

Line 128:  

Reviewer:  an edge image or the edge image

Line 131:  .. To the best of our knowledge

Reviewer:  alright, but this is already mentioned and I don't find it useful to keep repeating it!

Figure 3:  

Reviewer: here the concept of simultaneous scans is impractical, the TLS instruments are expensive in terms of cost, set up and heaviness. So I don’t think it’s a good idea. Furthermore, if the PCEs are not regular rectangular shapes, then the assumption in fig. 3 is not sufficient

Line 273:  based on RGB values

Reviewer:  the laser scan data are not RGB so how come? Is there a camera integrated?

Line 272:  the image is Z axis direction

Reviewer:  So you mean a TOP view

Line 275:  of the image N

Reviewer:  N is the image or the grid? Please rewrite

line 277:  in an grid

Reviewer: bad English

Line 276:  

Reviewer: d may be not a direction, but a step length or grid sampling distance

Line 281:  As the RGB mapping..

Reviewer:  I still think RGB mapping is a wrong terminology because you don’t have RGB values in the first place but only intensity so you mean intensity images?!

Line 288:  

Reviewer: this is just a guess! The value of N is related to the density of the scanning points and the size of the created images that you didn’t mention to us yet!

Line 291:  called as opening and closing by reconstruction

Reviewer:  So, your images are binaries and not RGB?

Line 292:  a small filter size

Reviewer: this is vague. You need to quantify the filter size and your BW images size as well

Equation 3:  

Reviewer: still confusing how you get the RGB info from laser scanning. You used integrated camera/images to add the texture? If yes, then why not using the images themselves and adding the depth info RGBD?.

line 383:  scanning environment s in..

Reviewer: environments?

line 429:   the 4-points congruent sets..

Reviewer:   unclear if this 4PCS is manual or automated?

Line 455:  efficacy

Reviewer: efficiency

Line 460:  

Reviewer: so in general based on your talk, we should use the grayscale scans to guarantee efficient data for all indoor and outdoor scenes.

Line 576:  approach performs well in..

Reviewer: this is not very scientific expression!

Line 578:  approach performs well in..

Reviewer: again here is a confusion!  few sentences before you said: based on RGB values of laser scans .. and now you repeat that gray scale scan is recommended!!!

Recommendations/future work:   

Reviewer: did you consider to use advanced machine learning/deep learning for the future segmentation and recognition?

Kind regards,

Author Response

Response to Reviewer 1 Comments

Point 1: Line 17:… use a coarse matching and a precise matching

Reviewer: may be coarse better replaced by and a fine matching

Response 1: Thank you for the suggestion, we have replaced ‘precise’ by ‘fine’ in this manuscript.

Point 2: Line 19:… outdoor laser scan data

Reviewer:  since you mentioned a specific number of millions or ten million, then I prefer to say: laser scan points

Response 2: Thank you, we have changed ‘laser scan data’ to ‘laser scan points’ as suggested.

Point 3: Line 22:  efficacy

Reviewer:  you mean efficiency!?

Response 3: Sorry, we have replaced this term by ‘effectiveness and efficiency’ to make the sentence clear.

Point 4: Line 22:  involved in this study

Reviewer: you mentioned experimental studies so more than one study but then here you say this study!

I expect to write in these studies,

or you mean in this paper!

Response 4: Thanks for the suggestion, we have revised “…in this study” to “in this paper” as suggested.

Point 5: Line 54:  affected by the quality of the images and the lighting conditions

Reviewer:  still not convincing to me why the image based is not suitable as much as TLS!

with the current dense matching and reconstruction even with open source tools, we can achieve very reliable dense point clouds of the PCEs especially if we also considered the texture in the segmentation which is not available in the raw laser data.

Response 5: Thank you very much for this comment. We fully agree with that the image-based methods are more computational efficient in contrast to the TLS-based methods. For scenarios such as indoor environment, generating dense point cloud data from images is a good choice when the accuracy can be guaranteed, such as concrete damage detection. However, due to environmental conditions, the obtained image could have poor quality, which may affect the accuracy of PCE dimension estimation.

In reality, the quality inspection environment for PCEs is usually an open outdoor environment. Therefore, using the TLS will make it easier to acquire the PCE data. In addition, 3D coordinates from laser scan data can facilitate the calculation of the main dimensions of PCEs. Currently, some commercial TLSs have built-in cameras that can get the RGB information for scanned objects. This also provides more information of PCEs for further quality inspection. Therefore, in contrast to the image-based methods, it is desirable to use the TLS-based methods for the quality inspection of multiple PCEs. For examples, Rabbani and Heuvel[1] believed that image-based methods, particularly when not implementing dense stereo-vision, will always remain limited because they aim at extracting dense 3D information from 2D images. Bosché[2] explained that no results have yet been published demonstrating the feasibility and reliably of automatically analyzing this sparse 3D data to infer such information as progress or dimensional quality.

[1] T. Rabbani, F. van den Heuvel, 3D industrial reconstruction by fitting CSG models to a combination of images and point clouds, in: International Archives of the Photogrammetry, Remote Sensing and Spatial Information Sciences (ISPRS), vol. XXXV-B5, Istanbul, Turkey, July 12–13, 2004, pp. 7–12.

[2] F. Bosche, Automated recognition of 3D CAD model objects in laser scans and calculation of as-built dimensions for dimensional compliance control in construction, Adv. Eng. Inform. 24 (2010) 107–118

Point 6: Line 58:  0.3_0.4mm within 10m…

Reviewer: you need first to clarify the required dimensional accuracy of the PCEs according to international standards and then you discuss the accuracy that can be attained by TLS or images,

Response 6: Thank you for the suggestion. We have added a new Table in Page 2, which shows the required dimensional accuracy of the conventional PCEs.                                                

Point 7: Line 99:  into 2D images, and perform..

Reviewer: here I am running into a conflict!! You mentioned that image based techniques are less sufficient compared with the TLS data, but now I am surprised that you are projecting the TLS data back to 2D images, so why not using images in the first place? You need to carefully clarify in the text

Response 7: We apologize for the possible confusion caused. We would like to clarify that as pointed out by the reviewer, image methods are computational efficient in contrast to TLS method approaches. In the proposed approach, we thus employed image-based method as one component for data segmentation, to speed up the whole PCE inspection procedure.

We have also added a description of the scanner used in this paper in Line 62-66 and Line 104-105. In this paper, we use the laser scan data containing 3D coordinates and RGB values. Therefore, converting 3D data into 2D images is to speed up the preliminary segmentation.

‘Usually, laser scan data obtained by commercial laser scanners contains not only the distance measurement of scanned objects, but also the red, green and blue (RGB) values of each point. Each point is represented as a six-dimensional array (x, y, z, r, g, b). These RGB values are calculated by the reflected signals from the scanned objects.’

‘Keeping the above in mind, in this paper, we choose the TLS with built-in camera to scan these PCEs.’

In addition, we have added the introduction of the color data and grayscale data obtained by the scanner used in this study in Line 278-280, which is given below:

‘It is worth noting that the color data obtained by the TLS has different values in the r, g, b dimensions, but the grayscale data has the same value in these three dimensions. In other words, the grayscale data can be also regarded as a special type of color data.’

Point 8: Line 128:

Reviewer:  an edge image or the edge image

Response 8: Thanks, we have corrected to ‘an edge image’ as suggested.

Point 9: Line 131:  .. To the best of our knowledge

Reviewer:  alright, but this is already mentioned and I don't find it useful to keep repeating it!

Response 9: We apologize for the repetition, we have deleted the repeated statement in the manuscript accordingly.

Point 10: Figure 3:  

Reviewer: here the concept of simultaneous scans is impractical, the TLS instruments are expensive in terms of cost, set up and heaviness. So I don’t think it’s a good idea. Furthermore, if the PCEs are not regular rectangular shapes, then the assumption in fig. 3 is not sufficient

Response 10: Thanks for the comments. We fully agree with the reviewer that TLS instruments are expensive in term of cost, set up and heaviness. This is exactly one of the motivation that we propose to scan multiple PCEs simultaneously. In particular, we would like to clarify that in the literature, PCEs are typically scanned in a sequential manner, which is time consuming. In the proposed method, we can save more time to obtain the TLS data of multiple PCEs. Here, we give an example to illustrate the comparison between the existing sequential based scanning and the proposed method, which are shown in the table below:

The number of scanned PCEs

The number of scans

Scanning time(s)

The sequential based   scanning

3

12

2004

The simultaneous scanning

3

8

1336

Fig.3 shows an illustrative example. The PCEs are not represented as regular rectangular shapes. For the irregular PCEs, we can also scan multiple PCEs simultaneously. Certainly, we will also carry out more detailed data collection.

Point 11: Line 273:  based on RGB values

Reviewer:  the laser scan data are not RGB so how come? Is there a camera integrated?

Response 11: Thank you for your comment. Yes, there are some commercial laser scanners with built-in cameras, which can convert reflected signals to RGB values and output them together with 3D coordinates.  The description of these scanner is given in Line 248-250 of the manuscript, which is given below:

‘Since the 3D coordinates and color information of the scanned object could be obtained by using a laser scanner with a built-in camera, the acquisition of color information provided another way to process the scanned PCE data. Wang et al. developed …’

Now, we have added a description of the scanner which can obtain RGB values, in Line 62-66 of the manuscript, as follows:

‘On the other hand, taking advantage of the spacious environment of the storage yard, the high-tech field data acquisition system (e.g., 3D laser scanner) is employed to obtain the geometric information of PCEs, which has the advantages of good ranging error (typically 2~6mm at 50m), low ranging noise (typically 0.3~0.4mm within 10m) and high measurement speed (up to 976,000 points/second). Usually, laser scan data obtained by commercial laser scanners contains not only the distance measurement of scanned objects, but also the red, green and blue (RGB) values of each point. Each point is represented as a six dimensional array (x, y, z, r, g, b). These RGB values are calculated by the reflected signals from the scanned objects.’

In addition, a description of the scanner used in this paper has been added in Line 104-105 as follows:

‘Keeping the above in mind, in this paper, we choose the TLS with built-in camera to scan these PCEs. Instead of dealing with the complex laser scan data directly…’

Point 12: Line 272:  the image is Z axis direction

Reviewer:  So you mean a TOP view

Response 12: We apologize for the confusion caused. The mapping image is not a TOP view image. The column of the image is the Z direction in the 3D data, and we have revised the sentence in Line 281-282, which is as follows:

‘The row of the image is the X or Y axis direction in the laser scan data, and the column of the image is the Z axis direction in the 3D data.’

Point 13: Line 275:  of the image N

Reviewer:  N is the image or the grid? Please rewrite

Response 13: We have rewritten this sentence (Line 282-286) to make the meaning of N clear. Please see below:

‘Taking account of the mapping efficiency, the same number of grids N is selected for both the row and column of the mapping image in the proposed approach. For instance, the following is to explain the use of YZ plane to perform RGB mapping. The number of girds N is an input parameter that needs to be determined initially. Then, the distance corresponding to the image grid can be obtained.’

Point 14: Line 277:  in an grid

Reviewer: bad English

Response 14: Thanks for point this out, we have corrected the typo as ’in a grid’ accordingly.

Point 15: Line 276:  

Reviewer: d may be not a direction, but a step length or grid sampling distance

Response 15: Thanks for your comment, we have made correction in Line 286-287 as follows:

‘In the horizontal direction and vertical direction, the gird distance dx and dz are calculated by Eq.1 and Eq.2, respectively.’

Point 16: Line 281:  As the RGB mapping..

Reviewer:  I still think RGB mapping is a wrong terminology because you don’t have RGB values in the first place but only intensity so you mean intensity images?!

Response 16: Many thanks for the comments. As the laser scan data used in this paper contains RGB values, we choose the term ‘RGB mapping’ here. Without loss of generality, both intensity and grayscale data can also be used in this step, please refer to Point 7 and 11 above for more details.

Point 17: Line 288:  

Reviewer: this is just a guess! The value of N is related to the density of the scanning points and the size of the created images that you didn’t mention to us yet!

Response 17: Thanks for your comment, we have added two images in the Figure 5 to show more test results.  Further, we have also discussed two criterions to configure N in the Line 297-300 of the manuscript, which is given below:

‘There are two criterions to select N:(1) Mapping images require enough pixels to ensure the image quality, (2) N should not be too large to reduce the spots in the mapping images and the running time.’

In addition, we have conducted more empirical studies using different N. The relationship between N and running time t is shown in the following figure. The configuration of N should be considered both the image quality and the running time indicated in the two criterions given above.

Point 18: Line 291:  called as opening and closing by reconstruction

Reviewer:  So, your images are binaries and not RGB?

Response 18: Thanks for the comment. The image is RGB image. When we obtain the data with RGB values, we can generate three images based on r, g, b value respectively. We can simply filter these images before proceeding to the next step. Then, the images for the next step are RGB images that consists of the processed r, g, b image.

Point 19: Line 292:  a small filter size

Reviewer: this is vague. You need to quantify the filter size and your BW images size as well

Response 19: Thank you for your comment. We have added the specific filter size (e.g. 3x3) for each image in Line 315 as suggested.

Point 20: Equation 3:  

Reviewer: still confusing how you get the RGB info from laser scanning. You used integrated camera/images to add the texture? If yes, then why not using the images themselves and adding the depth info RGBD?.

Response 20: Thank you for your comment. We scan the PCEs by using the TLS with a built-in camera. The data obtained by the TLS includes the 3D coordinates and RGB values. Then, we convert 3D data into 2D images based on the r, g, b value respectively. These three images can be converted into the RGB image. The image methods used in this paper aim to speed up the segmentation of laser scan data. For the quality inspection of PCEs, we want to obtain accurate coordinate information. In fact, we agree with that RGBD data is also a good choice when the measurement accuracy can meet the requirements of the specifications, but it is not as intuitive as the 3D coordinates of laser scan data for high-precision dimension calculations. Please refer to Point 5, 7, 11, 16 and 18 above for more details.

Point 21: line 383:  scanning environment s in..

Reviewer: environments?

Response 21: We apologize for the typo, and we have revised it accordingly.

Point 22: line 429:   the 4-points congruent sets..

Reviewer:   unclear if this 4PCS is manual or automated?

Response 22: Thanks for the comments. We have added a description of using the 4PCS to improve the success rate in Line 447-450 of the manuscript (please see below). The 4PCS used in this paper is based on RANSAC algorithm. By combining the max leverage sampling algorithm, we are able to achieve automatic coarse matching of PCE data with corresponding BIM model data.

‘In addition, by setting the minimum distance of the four randomly selected matching points (e.g., more than 0.5m), the success rate of the coarse matching can be improved. Based on the concept of RANSAC algorithm, the matching result with higher coarse matching accuracy will be automatically output.’

Point 23: Line 455:  efficacy

Reviewer: efficiency

Response 23: Thanks for your comment, and we have replaced this word by ‘effectiveness and efficiency’.

Point 24: Line 460:  

Reviewer: so in general based on your talk, we should use the grayscale scans to guarantee efficient data for all indoor and outdoor scenes.

Response 24: We apologize for the possible confusion caused. We agree that the color scans can provide the RGB information to aid in data segmentation or recognition. In the experimental studies, we use both color and grayscale data to verify the effectiveness of our proposed approach. We can verify the feasibility of our approach for grayscale data. Thereby, we have deleted this description.

Furthermore, we have added the introduction of the color data and grayscale data obtained by the scanner in this study in Line 278-280 as below:

‘It is worth noting that the color data obtained by the TLS has different values in the r, g, b dimensions, but the grayscale data has the same value in these three dimensions. In other words, the grayscale data can be also regarded as a special type of color data.’

Point 25: Line 576:  approach performs well in..

Reviewer: this is not very scientific expression!

Response 25: Thank you for pointing this out. We have rewritten this sentence in Line 594-596.

‘The proposed approach can segment all the PCE data, but it has poor performance in the background data segmentation.’

Point 26: Line 578:  approach performs well in..

Reviewer: again here is a confusion!  few sentences before you said: based on RGB values of laser scans .. and now you repeat that gray scale scan is recommended!!!

Response 26: We are grateful for this comment, and we have deleted this sentence that could be confused.

Point 27: Recommendations/future work:   

Reviewer: did you consider to use advanced machine learning/deep learning for the future segmentation and recognition?

Response 27: Many thanks for the excellent suggestion. As suggested, we are considering to further improve the recognition and segmentation performance of laser scan data by using deep learning techniques. We have added this in the discussion of future work.

‘In future work, the impact of these two aspects on the recognition results needs to be reduced. Further investigations are needed to extend its applicability to the simultaneous quality inspection of large batches of PCEs. We would like to use deep learning for an automatic segmentation and recognition of structural components in laser scan data of as-built buildings.’

Reviewer 2 Report

The work presents an interesting approach to segmenting PCEs automatically. The work is of general interest, the method is generally sound and the evaluation is essentially adequate, however, there are significant flaws. 

In Section 3.2.2 is not clear about whether or not the BIM model matching is automated; this needs to be made clear.

Check line 428 - the section number given appears to be incorrect, please check.

In Section 4, only grayscale scans are used for the storage yard evaluation (line 460). This suggests that the initial segmentation (RGB matching and clustering) mentioned in Section 3.1.3 , which uses RGB values, would not be applicable here. It is unclear why this method would be developed for RGB-based segmentation and not evaluated using RGB data.

Section 4.2 describes the segmentation results. Two issues arise here. Firstly, the RBNN is used as part of the segmentation process of the proposed approach, it should not be used for comparison. Secondly, one of the evaluation criteria is speed and the RBNN has been noted, in Section 3.1.4, as being efficient only for small amounts of data. Given the large amount of data generated by the scans, it is clear that RBNN will not be the most efficient comparative approach. My recommendation is that a state-of-the-art approach be used for evaluation.

The normalisation process described in line 492 is unclear.

Figure 19b shows the number of segments detected by both approaches. What are the actual number of segments? Which algorithm was more accurate?

The results in Figure 20 are not well described in the text and are, thus unclear. What do the numbers on the x-axis represent?

In lines 519-521 it is suggested that the initial filtering (which includes RGB-based processes) was responsible for the performance improvement in the proposed algorithm. This is problematic for a number of reasons. Firstly, not all evaluation was performed on colour images, why is there a performance difference in these circumstances? Secondly, would filtering before performing RBNN make it comparable to the proposed approach?

Author Response

Response to Reviewer 2 Comments

Point 1: In Section 3.2.2 is not clear about whether or not the BIM model matching is automated; this needs to be made clear.

Response 1: We apologize for the confusion caused. The BIM model matching is automated by combining the max leverage algorithm and 4PCS method. To make this clear, we have added the description of the matching step in Line 447-450 of the manuscript, as follows:

In addition, by setting the minimum distance of the four randomly selected matching points (e.g., more than 0.5m), the success rate of the coarse matching can be improved. Based on the concept of RANSAC algorithm, the matching result with higher coarse matching accuracy will be automatically output.

Point 2: Check line 428 - the section number given appears to be incorrect, please check.

Response 2: We apologize for the typo, we have corrected the section number as ‘Section 3.2.1’.

Point 3: In Section 4, only grayscale scans are used for the storage yard evaluation (line 460). This suggests that the initial segmentation (RGB matching and clustering) mentioned in Section 3.1.3 , which uses RGB values, would not be applicable here. It is unclear why this method would be developed for RGB-based segmentation and not evaluated using RGB data.

Response 3: Many thanks for the comments. Our method is developed for the laser scan data that consists of the 3D coordinates and RGB values. We have added a description of the scanner that can obtain RGB values in Line 62-66 as follows:

‘Usually, laser scan data obtained by commercial laser scanners contains not only the distance measurement of scanned objects, but also the red, green and blue (RGB) values of each point. Each point is represented as a six dimensional array (x, y, z, r, g, b). These RGB values are calculated by the reflected signals from the scanned objects.’

Actually, the RGB information is used as a tool to speed up laser scan data segmentation. Without loss of generality, both intensity and grayscale data can also be used in this step. we use both color and grayscale data to verify the effectiveness of our proposed approach. We can verify the feasibility of our approach for grayscale data. Thereby, in order to avoid the confusion, we have deleted this description.

Then, we have added the introduction of the color data and grayscale data obtained by the scanner in this study in Line 278-280 as below:

‘It is worth noting that the color data obtained by the TLS has different values in the r, g, b dimensions, but the grayscale data has the same value in these three dimensions. In other words, the grayscale data can be also regarded as a special type of color data.’

Point4: Section 4.2 describes the segmentation results. Two issues arise here. Firstly, the RBNN is used as part of the segmentation process of the proposed approach, it should not be used for comparison. Secondly, one of the evaluation criteria is speed and the RBNN has been noted, in Section 3.1.4, as being efficient only for small amounts of data. Given the large amount of data generated by the scans, it is clear that RBNN will not be the most efficient comparative approach. My recommendation is that a state-of-the-art approach be used for evaluation.

Response 4: Thanks very much for your valuable suggestions. The RBNN algorithm represents the graph-based method, here we control the same parameters to compare with the RBNN algorithm in order to highlight the improvement of image processing on 3D laser scan data. Although the RBNN algorithm was often used to process indoor point cloud data, but it is also an efficient graph-based method. Further, to the best of our knowledge, there is no existing work conducted on the PCE laser scan data with such a large scale. Therefore, there is no source codes available online for comparison. We thank the reviewer for this suggestion and seek the understanding from the reviewer.

In addition, in the next work, we will pay more attention to the use of deep learning methods to segment the PCE data. We have added the statement in the discussion of future work in Line 603-605 as follows:

We would like to use deep learning for an automatic segmentation and recognition of structural components in laser scan data of as-built buildings.

Point5: The normalisation process described in line 492 is unclear.

Response 5: Thanks for your comment, to make this clear, we have rewritten this description in Line 506-507 as follows:

‘Finally, in each experimental data, these Ers are normalized based on the maximum error rate which is calculated by the proposed approach and the RBNN algorithm for all segmented PCE data.’

Point6: Figure 19b shows the number of segments detected by both approaches. What are the actual number of segments? Which algorithm was more accurate?

Response 6:  Thank you for the comment. We would like to clarify that the goal of segmentation step is to separate all PCE data from the laser scan data. The actual number of segments is the number of PCEs. Both of two methods can obtain the segments of PCE data, we have added the figure, which is the manual segmentation result of the last experimental data GSD-3. We have rewritten the description of these result on Line 519-530.

‘However, the proposed approach fails to have good performance in the segmentation of background data, as shown in Figure 17(a) and Figure 19(b). Meanwhile, the RBNN algorithm has the under-segmented results for the background data. The numbers of segments obtained by the proposed approach are more than those obtained by the RBNN algorithm, considering the fact that the proposed approach does not consider the segments whose centroid distance are more than 5m. Nonetheless, the background data is not the focus of this study. This issue can be solved by scanning in a controlled manner like the second kind of scanning environment, as shown in the fifth row in Figure 17. Figure 20 shows the manual segmentation result of GSD-3 for PCE data. There are 17 segments selected manually in this experimental data. Due to the existence of filtering operation, the segmentation result of the proposed approach is better in line with the manual segmentation result in contrast to the RBNN algorithm according to Figure17 and19(b).’

Point7: The results in Figure 20 are not well described in the text and are, thus unclear. What do the numbers on the x-axis represent?

Response 7: We apologize for the missing of necessary information. The numbers represent the number of PCE data shown in Figure 18.

We have also added the description of these figures in Line 531-533, which is given below:

‘Figure 21 presents the comparison of normalized segmentation errors of the two methods on the PCE data in each experimental data. In these histograms, 1 represents the result with the largest error.’

We have added the description of the X axis in Line 507-509 as follows:

‘The error evaluation results are shown in Figure 21, in which the numbers on the X axis represent the number of PCE data shown in Figure18.’

Point8: In lines 519-521 it is suggested that the initial filtering (which includes RGB-based processes) was responsible for the performance improvement in the proposed algorithm. This is problematic for a number of reasons. Firstly, not all evaluation was performed on colour images, why is there a performance difference in these circumstances? Secondly, would filtering before performing RBNN make it comparable to the proposed approach?

Response 8: Thanks for the comment. Our proposed method is developed for the laser scan data which has the six dimensional data as (x, y, z, r, g, b). We have added the introduction of the color data and the grayscale data that obtained by the scanner used in this study in Line 278-280, which is given as follows:

‘It is worth noting that the color data obtained by the TLS has different values in the r, g, b dimensions, but the grayscale data has the same value in these three dimensions. In other words, the grayscale data can be also regarded as a special type of color data.’

Therefore, the proposed method can deal with the grayscale data. If the data is filtered before performing RBNN algorithm, the PCE segmentation effect will be improved. However, the processing speed may still be slower than the method based on image segmentation.

Reviewer 3 Report

First of all, I'd like to thank the authors for their efforts and the valuable work. The content is well organized and presented. The language is fine (just minor editions). However, I have some comments and recommendations.

Comments:

1- One of the most important motivations of your work is time saving. You also mentioned that the proposed approach saves more than 2 times with respect to running time of the RBNN algorithm. However, the is no clue in the experiments that supports the time saving conclusions.In other words, the experiments should illustrates the time consumption for certain number of PCEs with the proposed and RBNN algorithms.

2-  Title of table 6: language should be revised (single or plural).

3- Figures should be inserted after (not before) the corresponding paragraph which contains its caption in the context

Recommendation:

In RGB mapping, N is critical. You determined N based on some mapping tests. However, this selection is not valid at different environments. I strongly recommend to use/find a criterion to select the appropriate N.

Author Response

Response to Reviewer 3 Comments

Point 1: One of the most important motivations of your work is time saving. You also mentioned that the proposed approach saves more than 2 times with respect to running time of the RBNN algorithm. However, there is no clue in the experiments that supports the time saving conclusions.In other words, the experiments should illustrates the time consumption for certain number of PCEs with the proposed and RBNN algorithms.

Response 1: Thank you for your comment, we have shown the comparison of running time between the proposed approach and the RBNN algorithm in Fig.19(a).

The discussion of this figure is given in Line 517-520 as follows:

‘As shown in Figure 19(a), the proposed approach in this study provides faster segmentation because the image processing technology reduces the data processing time. The time spent by the RBNN algorithm is about 3 times of that the spent by the proposed approach. It is demonstrated that the proposed approach improves the processing speed for large data volumes.’

Point 2: Title of table 6: language should be revised (single or plural).

Response 2: Thanks for the comments, we have rewritten the title of the table, which is given as below:

‘The evaluation value DOC of the similar segment in the experiment of PCE amount.’

Point 3: Figures should be inserted after (not before) the corresponding paragraph which contains its caption in the context

Response 3: We are grateful for your comment, we have adjusted these figures to the appropriate place as suggested.

Point 4: Recommendation:

In RGB mapping, N is critical. You determined N based on some mapping tests. However, this selection is not valid at different environments. I strongly recommend to use/find a criterion to select the appropriate N.

Response 4: Thank you for the suggestion. We have added two images in the Figure 5 to show more test results. Moreover, we have also discussed two criterions to configure N in Line 293-295 of the manuscript, which is given as follows:

‘There are two criterions to select N:(1) Mapping images require enough pixels to ensure the image quality, (2) N should not be too large to reduce the spots in the mapping images and the running time.’

In addition, we have conducted more empirical studies using different N. The relationship between N and running time t is shown in the following figure. The configuration of N should consider both image quality and the running time as indicated in the two criterions given above.
